

**Size distribution and coating thickness of black carbon from the Canadian oil sands operations**
Yuan Cheng[1], Shao-Meng Li[1,*], Mark Gordon[2], Peter Liu[1]
[1] Air Quality Research Division, Environment and Climate Change Canada, 4905 Dufferin Street,
Toronto, Ontario M3H 5T4, Canada
[2] Department of Earth and Space Science and Engineering, York University, 4700 Keele Street,
Toronto, Ontario M3J 1P3, Canada
* Corresponding author: shao-meng.li@canada.ca
**Abstract**
Black carbon (BC) plays an important role in the Earth's climate system. However, parameterization
of BC size and mixing state have not been well addressed in aerosol-climate models, introducing
substantial uncertainties into the estimation of radiative forcing by BC. In this study, we focused on
BC emissions from the massive oil sands (OS) industry in northern Alberta, based on an aircraft
campaign conducted over the Athabasca OS region in 2013. A total of 14 flights were made over the
OS source area, in which the aircraft was typically flown in a 4- or 5-sided polygon pattern along
flight tracks encircling an OS facility. Another 3 flights were performed downwind of the OS source
area, each of which involved at least three intercepting locations where the well-mixed OS plume was
measured along flight tracks perpendicular to the wind direction. Comparable size distributions were
observed for refractory black carbon (rBC) over and downwind of the OS facilities, with rBC mass
median diameters (MMD) between ~ 135 and 145 nm that were characteristic of fresh urban
emissions. This MMD range corresponded to rBC number median diameters (NMD) of ~ 60−70 nm,
approximately 100% higher than the NMD settings in some aerosol-climate models. The typical in-
and out-of-plume segments of a flight, which had different rBC concentrations and photochemical
ages, showed consistent rBC size distributions. Moreover, rBC size distributions remained unchanged
at different downwind distances from the source area, suggesting that atmospheric aging would not
necessarily change rBC size distribution. However, aging indeed influenced rBC mixing state. Coating
thickness for rBC cores in the diameter range of 130−160 nm was nearly doubled within three hours
when the OS plume was transported over a distance of 90 km from the source area.





## 1. Introduction

Oil sands (OS), a type of unconventional petroleum deposit, are naturally occurring mixtures of bitumen (an extremely viscous form of crude oil), sand, water, and small amounts of other contaminants. The OS deposit in Alberta, Canada, is estimated to contain about 1.7 trillion barrels of bitumen. This deposit is distributed in the Athabasca, Cold Lake and Peace River regions, covering a total area of $\sim 1.42 \times 10^5$ km$^2$, of which about 10% can be recovered economically with existing technologies (Government of Alberta, 2009). Bitumen can be recovered in two ways, i.e., surface mining for the shallow reserves (e.g., less than 75 m below the surface) and using in situ technologies for the deeper deposits. Surface mining can be applied to an area of only 4800 km$^2$ area within the Athabasca region; and by 2013, about 19% of this surface minable area had been disturbed (Alberta Energy, 2017). As demand for crude oil continues to increase, oil production from the Alberta oil sands has experienced rapid expansion over the last decade, with total OS production doubling between 2004 (1.1 million barrels per day, with about 66% from surface mining) and 2014 (2.2 million barrels per day, with about 47% from surface mining) (Alberta Energy, 2016).

The massive OS industry in Alberta has raised substantial concerns on environmental impacts. For example, studies by Kelly et al. (2009, 2010) and Kurek et al. (2013) showed that the OS development contributed substantial amounts of organic (e.g., polycyclic aromatic hydrocarbons, PAHs) and inorganic (e.g., mercury, nickel, and thallium) pollutants to the Athabasca River watershed. Moreover, model simulations by Parajulee and Wania (2014) indicated that the Canadian National Pollutant Release Inventory (NPRI) likely underestimated PAHs emissions in the Athabasca OS region. Despite these studies, both the emissions and subsequent environmental impacts remain poorly understood for pollutants from the Alberta OS industry. To address this lack of understanding, an aircraft campaign was conducted with measurements of an extensive set of air pollutants over the Athabasca OS region in the summer of 2013. Using results from the campaign, Shephard et al. (2015) validated profiles of ammonia, carbon monoxide, formic acid, and methanol retrieved from the Tropospheric Emission Spectrometer (TES) satellite; Liggio et al. (2016, 2017) demonstrated the large OS surface mining facilities in Athabasca as a significant source of secondary organic aerosol (SOA)


and gaseous organic acids; and Li et al. (2017) identified the surface mining facilities as a greater
source of volatile organic compounds (VOCs) than previously realized.

In addition to gaseous pollutants and SOA, another focus of the 2013 aircraft campaign is black

carbon (BC) emissions from the surface mining facilities and its transport downwind. BC is a distinct
type of carbonaceous material formed during incomplete combustion of fossil and biomass fuels,
which is strongly light-absorbing in the visible light spectral range, refractory, insoluble and typically
appears as chain-like aggregates consisting of fewer than 10 to several hundred carbon spherules
(Andreae and Gelencsér, 2006; Bond et al., 2013; Petzold et al., 2013; Buseck et al., 2014). BC plays
a unique and important role in the Earth's climate system as an effective absorber of solar radiation. It
has relatively short atmospheric residence times but can exert a strong warming effect on global and
regional climate (Ramanathan and Carmichael, 2008; Bond et al., 2013; Myhre et al., 2013).
Therefore, BC emission reduction has long been considered as an important near-term climate
mitigation target. However, each step along the way between source and environmental effect of BC is
complex. For example, anthropogenic BC emissions and the resulting temporal and spatial variations
of BC, which can be simulated by chemical transport models, remain highly uncertain (Samset et al.,
2014). On the other hand, parameterization of BC size and mixing state has not been well addressed in
state-of-the-art radiative transfer models (Morgenstern et al., 2017). Both factors are recognized as
important sources of uncertainties in the estimate of climate forcing by BC (IPCC, 2013).

For large-scale industrial activities such as the OS surface mining operations in Athabasca, key

concerns regarding BC include (but are not limited to) the magnitude of BC emitted into the
atmosphere, size distribution and mixing state of the freshly emitted BC particles, evolution of the BC
particles including their size, mixing state and optical properties as the OS plumes are transported
downwind, and BC deposition. In this study, a total of 17 flights conducted during the 2013 aircraft
campaign were investigated to characterize BC emissions from six major OS surface mining facilities
in the Athabasca region, with focuses on the evolution of BC size distribution and mixing state.
Airborne BC measurements were performed by a Single Particle Soot Photometer (SP2). BC mass and
number size distributions were determined and compared not only for different facilities but also for



different downwind distances. It is commonly believed that BC cores in aged air masses are larger in
size compared with those in fresh emissions (e.g., Moteki et al., 2007; McMeeking et al., 2010). Here
we demonstrate that this is not necessarily the case. BC mixing state was estimated by coating
thickness retrieved from the SP2, based on which the influences of photochemical aging were
illustrated. Limitations of using this coating thickness to represent BC mixing state were also
discussed. These results can provide insights into the evolution of BC aerosol in the real atmosphere.
**2. Methods**
**2.1 Aircraft campaign**
The aircraft campaign was conducted over the Athabasca OS region in northern Alberta between
August 13 and September 7, 2013 in support of the Joint Canada-Alberta Implementation Plan for Oil
Sands Monitoring (JOSM). Using a suite of state-of-the-art instruments installed aboard the National
Research Council Institute for Aerospace Research Convair-580 research aircraft, an extensive set of
air pollutants (including both gaseous and particulate species) were determined with high time
resolutions (Gordon et al., 2015; Liggio et al., 2016; Li et al., 2017). During this campaign, 22 flights
were made over the Athabasca OS region, for a total of about 84 hours. These flights were designed (1)
to quantify emissions of air pollutants from six major OS surface mining facilities including Syncrude
Mildred Lake (SML), Suncor Energy OSG (SUN), Canadian Natural Resources Limited Horizon
(CNRL), Shell Albian and Jackpine (SAJ), Syncrude Aurora (SAU), and Imperial Kearl Lake (IKL),
and (2) to determine atmospheric evolution of the primary pollutants. The details of the measurements,
the flight patterns, and objectives of the flights were described in detail by Liggio et al. (2016) and Li
et al. (2017). In 14 flights for emission quantitation, the aircraft was typically flown in a 4- or 5-sided
polygon pattern encircling an OS surface mining facility, with level flight tracks at 8−10 altitudes
increasing from 150 to 1370 m above ground; these level flight tracks were stacked along the sides of
the polygon to form a virtual box encasing the facility (Figure 1a). Repeated emission flights were
made over SML, SUN, CNRL, and SAJ, whereas single flights were made over SAU and IKL.
Three flights were designed to study transformation of air pollutants emitted from the OS surface
mining facilities. They were conducted in a Lagrangian pattern such that the same OS plume was



sampled at different time intervals (approximately 1 hour apart) as it was transported downwind from
the source area (Figure 1b). Real-time wind speed and direction measurements were used to guide the
intercepting locations. The first intercepting locations were chosen at about 1 hour downwind of the
majority of the OS facilities so that the emitted air pollutants were well mixed and merged into large
plumes. At each intercepting position, the aircraft was flown along level flight tracks perpendicular to
the wind direction at multiple altitudes; then these level flight tracks were stacked vertically to create a
virtual screen downwind of the OS source area. At least three screens were created for each
transformation flight, without industrial emissions in between.
**2.2 BC measurements by the SP2**

A Single Particle Soot Photometer (SP2; Droplet Measurement Technologies Inc., Boulder, CO,

USA) was used to measure the refractory black carbon (rBC) cores on a particle-by-particle basis
based on incandescent light emitted from heated rBC cores when they cross and absorb energy from a
laser beam (Stephens et al., 2003; Baumgardner et al., 2004; Schwarz et al., 2006; Moteki and Kondo,
2010; Laborde et al., 2012a). The SP2 used in this study detected single particle rBC cores in the mass
range of ~ 0.3−16 fg, based on the calibration using regal black particles (Cappa et al., 2012). To
account for the rBC cores outside this detection range, a lognormal fit was applied to the measured
rBC size distribution and then extrapolated over 10−1000 nm (Schwarz et al., 2006). Here the rBC
size refers to the mass equivalent diameter ($D_{MEV}$) calculated as $\left[ (6 \times m)/(\rho \times \pi) \right]^{1/3}$, where $m$ and $\rho$
are the mass and density of the rBC core, respectively. The value of $\rho$ was assumed to be 1.8 g/cm$^3$,
which corresponds to the median $\rho$ value recommended by Bond and Bergstrom (2006). Using this $\rho$
value, the rBC detection range could be converted to ~ 70−260 nm in terms of $D_{MEV}$. For either rBC
mass or number concentration, a scaling factor ($F_{rBC}$) was calculated as $I_{whole}/I_{detected}$, where $I_{whole}$
indicates the integral of the lognormal fitting curve from 10 nm to 1000 nm, and $I_{detected}$ indicates the
integral of the curve from 70 nm to 260 nm. Subsequently, the final rBC concentration could be
determined as $F_{rBC} \times C_{detected}$, where $C_{detected}$ is the detected rBC concentration (either mass or number)
measured by the SP2. All the rBC concentrations involved in this paper have been scaled by $F_{rBC}$.



135 In addition to emitting incandescent radiation, rBC containing particles also scatter light when

136 passing through the laser beam of the SP2. Coating thicknesses on rBC cores ($T_{coating}$, in nm) can be

137 retrieved from the scattering signals on a particle-by-particle basis, using Mie theory calculation with

138 a series of assumptions (Schwarz et al., 2008a, b; Laborde et al., 2012b). To calculate $T_{coating}$ for an

139 rBC containing particle, the internally mixed particle needs to be idealized as a two-component sphere

140 with a concentric core-shell morphology. In this study, the rBC core was assumed to have a complex

141 refractive index of $2.26 - 1.26i$, which was initially suggested by Moteki et al. (2010) and

142 subsequently confirmed by Taylor et al. (2015). The coating material on a rBC core was assumed to

143 have a complex refractive index of $1.5 - 0i$, which is representative of the corresponding values

144 determined for inorganic salts (e.g., ammonium sulfate) and secondary organic aerosol (Schnaiter et

145 al., 2005; Lambe et al., 2013). The core size was held fixed at $D_{MEV}$ of the rBC core, whereas the

146 diameter of the whole particle was varied in the Mie calculation until the modeled scattering cross

147 section matched the measurement. Measured scattering cross section was determined by a leading-

148 edge-only (LEO) fit to the recorded scattering signal (Gao et al., 2007). Finally, $T_{coating}$ was calculated

149 as the difference between the radii of the whole particle and the rBC core.

150 **3. Results and Discussion**

151 **3.1 rBC size distributions over the OS source region: facility-integrated results**

152 For each flight, the measured masses of the individual rBC cores over the entire flight were first

153 grouped into different size bins and then fitted by a lognormal curve:

$$\frac{\mathrm{d}m}{\mathrm{d}\log D_{MEV}} = A_{mass} \times \exp\left\{ 0 - \left[ \frac{\ln\left(D_{MEV}/X_{1,\,mass}\right)}{X_{2,\,mass}} \right]^2 \right\} \tag{1}$$

155 where $A_{mass}$, $X_{1,\,mass}$ and $X_{2,\,mass}$ are the fitting parameters. The fitting parameter $X_{1,\,mass}$ will be termed

156 the mass median diameter (MMD), and the fitting parameter $X_{2,\,mass}$ will be loosely referred to as the

157 mass distribution width. As can be seen from Equation (1), $A_{mass}$ is proportional to the absolute value

158 of rBC mass concentration and thus it is unimportant for describing the shape of a lognormal curve.

159 This is particularly the case for comparison of rBC size distributions among different OS facilities. It



should also be noted that the mass-based scaling factor ($F_{rBC, mass}$), which accounts for the rBC masses
outside the SP2's detection range, is independent of $A_{mass}$. Therefore, $A_{mass}$ will not be further
discussed in rBC size distribution. Similarly, rBC number-size distribution could be expressed as:
$$\frac{dN}{d\log D_{MEV}} = A_{number} \times \exp\left\{0 - \left[\frac{\ln\left(D_{MEV}/X_{1,\,number}\right)}{X_{2,\,number}}\right]^2\right\}$$
(2)

where $A_{number}$, $X_{1,\,number}$ and $X_{2,\,number}$ are the fitting parameters. $X_{1,\,number}$ and $X_{2,\,number}$ will be termed
the number median diameter (NMD) and the number distribution width, respectively.
Mass and number size distributions of rBC are summarized in Figure 2 for the 14 emission
flights. As shown in Figure 2, the rBC MMD and NMD were typically in the range of 135−145 nm
and 60−70 nm, respectively, while both the mass and number distribution widths were approximately
0.7. Most of the rBC from the surface mining facilities were from the heavy diesel trucks used to
transport the mined oil sands ores to centralized locations in each facility for bitumen separation from
the sands. In most cases, rBC emissions from the six major OS surface mining facilities exhibited
similar size distributions. These rBC size distributions are comparable with those observed for urban
emissions and source (or near-source) samples representing different types of engine exhausts. For
example, (1) during an airborne measurement conducted as part of the CalNex 2010 campaign, rBC
MMD was estimated to be 122 nm over the Los Angeles Basin (Metcalf, et al., 2012); (2) rBC MMD
observed in the urban outflows were typically in the range of 140−180 nm, as evidenced by ground-
based measurement downwind of Tokyo (Shiraiwa et al., 2007), and by aircraft-based observations
over Texas (Schwarz et al., 2008a), California (Sahu et al., 2012) and western and northern Europe
(McMeeking et al., 2010); (3) when mainly impacted by traffic emissions, rBC MMD were about 100
and 120 nm for a suburban site in Paris (Laborde et al., 2013) and an urban site in London (Liu et al.,
2014), respectively; (4) rBC MMD measured at urban sites in Tokyo, Japan (Kondo et al., 2011b) and
Sacramento, CA (Cappa et al., 2012) were between 140 and 150 nm; (5) a laboratory study showed
that the MMD was about 125 nm for rBC cores emitted from a diesel car (Laborde et al., 2012b); (6) a
MMD of 126 nm was observed for rBC at the Cranfield airport in UK, attributable to aircraft engine



exhausts (McMeeking et al., 2010). Although not all of these studies determined rBC MMD and NMD
simultaneously, rBC NMD were typically in the range of ~ 60 to 70 nm for urban emissions
dominated by contributions from fossil fuel combustion (e.g., Schwarz et al., 2008a; Kondo et al.,
2011b; Metcalf, et al., 2012).

A comparison of rBC size distributions between this study and previous ones also suggests that

rBC cores emitted from fossil fuel combustion were smaller in size compared to those from biomass
burning. The rBC MMD and NMD measured in biomass burning plumes were typically around 200
and 140 nm, respectively, which were supported by airborne measurements over Texas (Schwarz et al.,
2008a), California (Sahu et al., 2012), Canada (Kondo et al., 2011a; Taylor et al., 2014) and the Arctic
(Kondo et al., 2011a). However, wet deposition could lead to a large decrease (e.g., as much as 50 nm)
in the MMD of rBC cores in biomass burning plumes (Taylor et al., 2014), suggesting that an rBC
MMD substantially smaller than 200 nm does not exclude the possibility of biomass burning emission
contributions.

Different assumptions have been made by aerosol-climate models for the size distribution of

black carbon. For example, the NMD of black carbon emitted by fossil fuel combustion were assumed
to be 30, 40 and 60 nm by Dentener et al. (2006; for AeroCom Phase I models), Heald et al. (2014; for
a radiative transfer model coupled with GEOS-Chem) and Stier et al. (2005; for the aerosol-climate
modelling system ECHAM5-HAM), respectively. According to the SP2 measurement results on rBC,
including those from the present study, a NMD of 60 nm would be a more appropriate input parameter
in the models for black carbon emissions from fossil fuel combustion.
**3.2 rBC size distributions over the OS source region: time-resolved results**

In addition to the facility-integrated results (Figure 2), log-normal fits were also applied to 2-min

intervals of rBC data derived from the SP2. Figure 3 and 4 show results from the emission flights
conducted for CNRL on August 26, 2013 (i.e., F_8/26) and for SUN on August 28, 2013 (i.e., F_8/28),
respectively. In both cases, the rBC mass and number size distributions did not exhibit major temporal
variations, despite the minor fluctuations observed during F_8/28. The stable rBC size distribution
within a flight can be more readily seen from Figure 5a, which indicates that the rBC MMD, mass



distribution width and therefore the mass-based scaling factor ($F_{rBC, mass}$) were independent of rBC
concentration. As shown in Figure 5a and Table 1, the variations of rBC MMD, mass distribution
width, and $F_{rBC, mass}$ were within 5% for F_8/26. Larger variations in rBC size distribution were
observed for F_8/28, but the variations in these three parameters were still within 10%. The variations
of rBC NMD, number distribution width, and number-based scaling factor ($F_{rBC, number}$) were also
within 10% for both F_8/26 and F_8/28 (Table 1).

The temporal variations of rBC concentration shown in Figure 3 and 4 were mainly driven by the

in- vs. out-of-plume differences. There were a sharp increase in rBC concentration when the aircraft
flew into a plume, whereas the rBC concentration deceased rapidly when the aircraft left the plume.
Therefore, the stable rBC size distributions observed for the emission flights, which were clearly
independent of rBC concentration (e.g., Figure 5a), mean negligible in- vs. out-of-plume differences
in rBC size distributions over the OS source region. The size distribution consistency for rBC is
observed regardless of the threshold rBC concentration used to distinguish the in- and out-of-plume
conditions, which is flight-dependent (e.g., ~ 0.1 $\mu g/m^3$ in terms of 2-min averaged rBC mass
concentration for F_8/26 as shown in Figure S1). The implications of consistent size distributions for
rBC near the sources are further discussed in Section 3.3 together with results from the transformation
flights.

In addition to rBC concentration, the in- and out-of-plume air masses had different

photochemical ages. Here a photochemical age is calculated as $-\log_{10}(NO_x/NO_y)$, where $NO_x$ is the
sum of nitrogen monoxide and nitrogen dioxide (i.e., $NO + NO_2$) and $NO_y$ refers to the total reactive
oxidized nitrogen compounds (Kleinman et al., 2008). Measurement of $NO_x$ and $NO_y$ during the
aircraft campaign has been described elsewhere (Liggio et al., 2016). As shown in Figure 5b, there
was a robust negative correlation between the rBC mass concentration and photochemical age, which
likely reflects the connection between air mass dilution and aging. Compared to the in-plume
segments of a flight, the out-of-plume ones were characterized by not only lower rBC concentrations
but also older photochemical ages. Given the clear dependence of rBC concentration on
photochemical age (Figure 5b) and the stable rBC size distribution across the whole rBC





concentration range observed within an emission flight (Figure 5a and Table 1), it could be inferred
that rBC size distribution was independent of photochemical age over the OS source region.

**3.3 rBC size distributions downwind of the OS source region**

Mass and number size distributions of rBC are shown in Figure 6 and 7, respectively, for the
transformation flight conducted on September 4, 2013 (i.e., F_9/4) which reached a downwind
distance of approximately 120 km (relative to the downwind edge of the OS source area; Figure 1b).
As can be seen from the time-resolved log-normal fitting results (Figure 6a and 7a), both the rBC
mass and number size distributions were fairly stable during F_9/4, without major temporal change
patterns. For the typical in- and out-of-plume conditions of F_9/4, the rBC MMD were $143.39 \pm 0.95$
and $141.56 \pm 1.19$ nm with mass distribution widths of $0.72 \pm 0.01$ and $0.71 \pm 0.02$, respectively
(Figure 6b); the rBC NMD were $70.65 \pm 0.42$ and $69.02 \pm 0.46$ nm with number distribution widths of
$0.68 \pm 0.01$ and $0.69 \pm 0.01$, respectively (Figure 7b). These rBC size distributions (Figure 6b and 7b)
were derived from the SP2 measurements performed on the various virtual screens, where the aircraft
was flown along level flight tracks (primarily at ~ 450 and 600 m) perpendicular to the wind direction.
For the level flight tracks, the typical in- and out-of-plume conditions (i.e., segments) were
distinguished by rBC concentration (Figure 8), i.e., the typical out-of-plume conditions were identified
by relatively low and constant rBC concentrations whereas the typical in-plume conditions were
characterized by sharp increases in rBC concentration above the out-of-plume level. In Figure 6b, the
rBC mass size distribution was scaled for the out-of-plume conditions to reveal their lower rBC
concentrations compared to the in-plume conditions (Figure 6d). When performing the scaling, the in-
plume rBC size distribution was used as a reference (i.e., kept unchanged). The out-of-plume rBC size
distribution was scaled to make the $I_{\text{out-of-plume, scaled}}$ to $I_{\text{in-plume}}$ ratio equal the $\text{rBC}_{\text{out-of-plume}}$ to $\text{rBC}_{\text{in-plume}}$
ratio, where the individual terms, in sequence, represent integral of the scaled out-of-plume rBC size
distribution curve, integral of the reference in-plume rBC size distribution curve, the average out-of-
plume rBC mass concentration ($54.22$ ng/m$^3$, derived from Figure 6d), and the average in-plume rBC
concentration ($207.93$ ng/m$^3$, derived from Figure 6d). In Figure 7b, the out-of-plume rBC number





size distribution was scaled in the same way. As can be seen from Figure 6b and 7b, the in- vs. out-of-
plume difference was negligible for rBC size distribution downwind of the OS region.

Photochemical ages were older for the out-of-plume conditions compared to the in-plume ones,

by ~ 0.3−0.5 in terms of $-\log_{10}(NO_x/NO_y)$ for different screens of F_9/4 (Figure 6e). Therefore, the
consistent rBC size distributions between the in- and out-of-plume conditions indicated that
photochemical age had little influence on rBC size distribution downwind of the OS region. This
conclusion was also strongly supported by the comparison of in-plume rBC size distributions among
different downwind distances. As the OS plume was transported downwind, the in-plume rBC
concentration decreased due to dilution (Figure 6d), from ~ 310 ng/m$^3$ for the first screen (screen #1)
to ~ 110 ng/m$^3$ for the fourth screen (screen #4); on the other hand, the in-plume photochemical age
$-\log_{10}(NO_x/NO_y)$ increased (Figure 6e), from ~ 0.1 for screen #1 to ~ 0.5 for screen #4. The last
screen (screen #5) did not differ largely from screen #4 with respect to either in-plume rBC
concentration or photochemical age, appearing to indicate that the dilution and aging processes had
slowed down or even stopped since screen #4. However, it should be noted that unlike the first four
screens, screen #5 did not captured the full OS plume, i.e., the plume edges were missed. Compared to
the central portion of the plume, the plume edges had lower rBC concentrations and older
photochemical ages. Therefore, the average rBC concentration and $-\log_{10}(NO_x/NO_y)$ could not be
compared directly between screen #5 and the first four screens, and consequently, results from screen
#5 were not involved in Figure 6d and 6e. Nonetheless, for all successive screens of F_9/4, the in-
plume rBC MMD and NMD were found to fall into a narrow range of 140−145 and 69−72 nm,
respectively, while both the mass and number distribution widths were about 0.7 (Figure 6c, 7c and 9).
In Figure 6c and 7c, rBC size distributions derived from successive screens were scaled to show the
decreases in rBC concentration caused by dilution, using the same approach as that described in detail
for Figure 6b. The scaling requires rBC concentration representative of the full plume and thus was
not performed for screen #5. A direct comparison of rBC size distributions between screen #5 and the
first four screens is provided by Figure 9. Figure 9 also demonstrates consistent in-plume rBC size
distributions among successive screens for the other two transformation flights that were conducted on





August 19 and September 5, 2013, respectively (i.e., F_8/19 and F_9/5), providing further solid
evidence for the negligible influence of atmospheric aging on rBC size distribution downwind of the
OS source region.

Previous studies conducted in remote areas (either ground- or aircraft-based) typically showed

rBC MMD between 200 and 220 nm (Shiraiwa et al., 2008; Liu et al., 2010; McMeeking et al., 2010;
Schwarz et al., 2010), substantially higher than those observed over urban areas (e.g., 122 nm over the
Los Angeles basin; Metcalf et al., 2012) or at urban locations (e.g., 146 nm in Tokyo, Japan; Kondo et
al., 2011b). Therefore, it has been commonly believed that rBC cores in aged air masses are larger
than those in fresh emissions. However, results from the present study indicate that this is not
necessarily the case. It is inferred that not all aging processes will change rBC size distribution and
instead, influences of aging on rBC size distribution depend on the presence of atmospheric processes
that can lead to increased rBC core mass and size in a single particle (e.g., evaporation of cloud
droplets containing multiple rBC particles). In this study, it appears that no such processes were at
play, and within the photochemical ages encountered, rBC core masses and sizes did not change.

In addition to the evolution of in-plume rBC concentration, Figure 6d shows that the out-of-

plume rBC concentration decreased until screen #3. This decrease was associated with an increase in
$-\log_{10}(NO_x/NO_y)$ for the out-of-plume conditions (Figure 6e). For screen #4, both the out-of-plume
rBC concentration and photochemical age were nearly the same as the respective values observed for
screen #3. Therefore, the out-of-plume conditions identified for screens #3 and #4 should be more
representative of the background. For screens #3 and #4, rBC size distributions agreed well between
the in- and out-of-plume conditions, within ± 3 nm in terms of MMD or NMD, indicating that the
background did not differ significantly from the OS emissions with respect to rBC size distribution.
Consistent in- and out-of-plume rBC size distributions observed at smaller downwind distances (i.e.,
for screens #1 and #2) and over the OS source area (i.e., for the emission flights) pointed to the same
conclusion, although the out-of-plume conditions in these cases were less representative of the
background. rBC cores in the background could be from the OS emissions and/or long-range
transported urban emissions that had not been influenced by atmospheric processes that can change



single particle rBC core size. These two kinds of emissions did not differ largely in rBC size
distribution (as discussed in section 3.1) and therefore they were difficult to be further distinguished
only by rBC size.

**3.4 Evolution of rBC mixing state**

A key step to retrieve coating thickness ($T_{coating}$) of an rBC containing particle from its scattering
signal is the LEO fit, which requires, at least, the leading edge of the scattering signal ($S_{leading-edge}$) can
be properly measured (Schwarz et al., 2008a, b; Laborde et al., 2012b; Liu et al., 2014). However, the
LEO fit cannot be performed when $S_{leading-edge}$ is outside the SP2's detection range of scattering
intensity; thus, $T_{coating}$ cannot be calculated for relatively small rBC cores with thin coatings (i.e., rBC
containing particles with $S_{leading-edge}$ below the lower detection limit of scattering intensity) or
relatively large rBC cores with thick coatings (i.e., rBC containing particles with $S_{leading-edge}$ above the
upper detection limit of scattering intensity) (Metcalf et al., 2012; Dahlkötter et al., 2014). This
limitation prohibits a direct comparison of $T_{coating}$ across all rBC cores with different sizes. In this
study, $T_{coating}$ was found to exhibit a decreasing trend with the increase in rBC $D_{MEV}$ for both the
transformation (Figure 10) and emission flights (Figure S2). This trend was primarily attributed to the
limitation that the detection range of $T_{coating}$ is rBC $D_{MEV}$ dependent, rather than indicating that
relatively small rBC cores were more thickly coated than larger cores.
Besides $T_{coating}$, the fraction of rBC cores that can be assigned a coating thickness ($F_{assigned}$, in %)
was also rBC $D_{MEV}$ dependent such that $F_{assigned}$ was found to be the highest (between ∼ 35−45%) for
rBC cores in the $D_{MEV}$ range of 130−160 nm (Figure 10 and S2). The rBC containing particles in this
$D_{MEV}$ range were selected for further discussions on $T_{coating}$ (their $T_{coating}$ will be specified as $T^*$), with
a focus on the evolution of rBC mixing state as the OS plumes were transported downwind. As shown
in Figure 11a for the transformation flight F_9/4, the in-plume $T^*$ exhibited an increasing trend with
the increase in downwind distance or transport time, e.g., from ∼ 22 nm for screen #1 to ∼ 41 nm for
screen #4. This trend is not surprising given the continuous formation of SOA during transport of the
OS plumes (Liggio et al., 2016). For rBC near the sources, $T^*$ was close to zero as observed from the
emission flights over the OS facilities. For example, $T^*$ was derived at ∼ 3 nm for F_9/3 (Figure S2).




These freshly emitted rBC cores grew a coating of ~ 20 nm thickness in the first hour after emission,
when the OS plume was transported from the sources in the OS facilities to the downwind edge of the
OS region.
$T*$ were found to be comparable between the in- and out-of-plume conditions for screen #1,
which were ~ 22 and 23 nm, respectively (Figure 11a). It is unlikely that the out-of-plume $T*$ could be
as low was ~ 23 nm, if the majority of the out-of-plume rBC cores were from long-range transport and
thus had an aging time of much longer than one hour. Therefore, the rBC cores observed in the out-of-
plume conditions should also be influenced by emissions in the oil sands region albeit at much lower
air concentrations compared to the plumes, such as from on road traffic that was not part of any oil
sands surface mining facility.
Compared to the in-plume conditions, the increase in $T*$ was smaller for the out-of-plume
conditions as the OS plume was further transported from screen #1 (Figure 11a) and moreover, the
out-of-plume $T*$ stopped increasing after screen #3 such that it was ~ 32 nm for both screens #3 and
#4. One explanation for the different evolution patterns of the in- and out-of-plume $T*$, which had
comparable initial values (i.e., those for screen #1), is the less effective formation of coating materials
(e.g., SOA and sulfate) for the out-of-plume conditions than in plumes, given that coating precursors
(volatile organic compounds and sulfur dioxide) were much more abundant in the plumes from which
fast formation of organic aerosols was observed (Liggio et al., 2016). As shown in Figure 11b, the in-
plume OA to rBC ratio exhibited a robust increasing trend with the increase in downwind distance
(e.g., by ~ 150% for screen #4 relative to screen #1), whereas the increase in OA to rBC ratio was less
significant for the out-of-plume condition (e.g., by only ~ 45% for screen #4 compared to screen #1)
which was negligible between screens #3 and #4.
We did not compare $T_{coating}$ measured in this study with results from previous ones due to the
following reasons. (1) The detection range of $T_{coating}$ and therefore the estimated $T_{coating}$ depend on the
SP2's detection range of scattering intensity, which could differ substantially among different SP2
instruments. This dependency indicates that different SP2 instruments might lead to different $T_{coating}$
estimates even for the same ensemble of rBC containing particles. (2) The detection range of $T_{coating}$



and therefore the estimated $T_{coating}$ also depend on the rBC core size (i.e., $D_{MEV}$). Quite different $D_{MEV}$
ranges have been used by previous studies to estimate $T_{coating}$, e.g., 190−210 nm by Schwarz et al.
(2008a, b) vs. 162−185 nm by Langridge et al. (2012), indicating that these $T_{coating}$ estimates are not
directly comparable. (3) The $T_{coating}$ estimates could be considerably influenced by uncertainties
introduced by the LEO fit. These uncertainties can be evaluated using non-rBC containing particles.
The scattering signals of non-rBC containing particles always have the shape of a full Gaussian curve,
since they will not evaporate or change in size when passing through the SP2's laser beam. Thus, for
non-rBC containing particles, the LEO fit should in principle lead to the same scattering amplitude or
the same optical size ($D_{optical}$) as that retrieved from a fit to the full scattering signal (i.e., the full-
Gaussian fit) (Gao et al., 2007). In this study (Figure 12), the LEO and full-Gaussian fits agreed within
approximately ± 15% in terms of $D_{optical}$ for non-rBC containing particles. Here the $D_{optical}$ was
calculated from the fitted scattering amplitudes, by assuming a complex refractive index of $1.5 - 0i$ for
non-rBC containing particles. $D_{optical}$ was used in Figure 12 to evaluate the agreement between the
LEO and full-Gaussian fits because it was more directly related to $T_{coating}$ compared to the scattering
amplitude. However, comparison of the LEO and full-Gaussian fits for the determination of $D_{optical}$ or
scattering amplitude was not presented in many previous publications involving $T_{coating}$. This is a
substantial concern because the LEO fit has been considered reliable as long as the LEO to full-
Gaussian ratios are relatively constant (not necessarily around 1.0) for the fitted scattering amplitudes
(e.g., Metcalf, et al., 2010). Since an agreement between the LEO and full-Gaussian fits was not
always required, previously reported $T_{coating}$ might be biased by the LEO-induced uncertainty to
different extents, adding to the difficulties in comparing $T_{coating}$ across studies.
**4. Conclusions**

An aircraft campaign was conducted over the Athabasca oil sands region in the summer of 2013,

during which the size distribution and coating thickness of refractive black carbon (rBC) cores were
studied as they were emitted from the sources and as they were transported downwind. The rBC size
distributions were found to be comparable at the six major OS surface mining facilities, typically with
mass median diameters (MMD) of 135–145 nm and number median diameters (NMD) of 60–70 nm



that were characteristic of fresh urban emissions dominated by contributions from fossil fuel
combustion. The results from the present study indicate that the size distributions assumed in some
aerosol-climate models for fossil fuel BC have NMD (e.g., 30 nm) which are likely too low by a factor
of 2 compared to the present results as well as previously reported values. An NMD of 60 nm would
be more appropriate for fossil fuel BC. The observed rBC size distributions were consistent not only
for the typical in- and out-of-plume segments of a flight (either emission or transformation) but also
for different downwind distances form the OS source area, indicating little dependence of the rBC size
distribution on atmospheric aging within 4 to 5 hours from the point of emission.

The coating thicknesses ($T_{coating}$) were retrieved for rBC containing particles from their scattering

signals, on a particle-by-particle basis. Mainly due to the SP2's limited detection range of scattering
intensity, $T_{coating}$ could not be calculated for all the detected rBC cores. The fraction of rBC cores that
can be assigned a coating thickness was found to be the highest but still lower than 50% for those in
the diameter range of 130−160 nm. It is not surprising that $T_{coating}$ increased as the OS plumes were
transported downwind, resulting from the formation mainly of secondary organic aerosols but also of
sulfate. Such coating increase with aging can significantly change the optical properties of the rBC
containing particles and hence their potentials for radiative forcing. Based on the present $T_{coating}$ results,
however, estimates of these effects are challenging, mainly because $T_{coating}$ was unknown for the
majority of the rBC containing particles.
**Acknowledgements**
We would like to thank the National Research Council of Canada flight crew of the Convair-580, the
technical support staff of the Air Quality Research Division, and Dr. Stewart Cober for the
management of the study. This project was supported by Environment and Climate Change Canada's
Climate and Clean Air Programme (CCAP) and the Canada-Alberta Joint Oil Sands Monitoring
program.

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





**Table 1.** Variations of the parameters derived from time-resolved lognormal fits to single-particle rBC data measured during F_8/26 and F_8/28. Variations are determined as relative standard deviations (RSD, in %).

|        | MMD  | Width$_{mass}$ | $F_{rBC, mass}$ | NMD  | Wdith$_{number}$ | $F_{rBC, number}$ |
|--------|------|----------------|-----------------|------|------------------|-------------------|
| F_8/26 | 1.46 | 4.42           | 2.82            | 4.48 | 5.30             | 4.07              |
| F_8/28 | 6.85 | 8.46           | 9.47            | 7.94 | 7.18             | 8.07              |


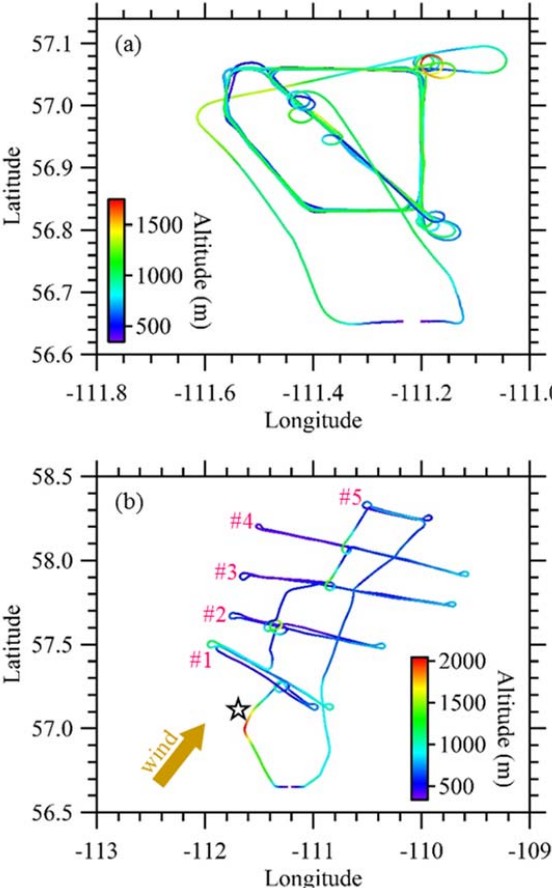

**Figure 1.** Examples of flight tracks for **(a)** emission and **(b)** transformation flights, which were flown on August 28 (F_8/28) and September 4 (F_9/4), 2013, respectively. F_8/28 was flown in a 5-sided polygon pattern, encircling the SUN facility. F_9/4 was conducted in a Lagrangian pattern, involving five virtual screens (#1 to #5) the first of which was located at the downwind edge of the OS source region. Distances between the successive flight screens during F_9/4 were approximately 30 km, whereas distance between the OS center (shown approximately by the open star) to screen#1 was also about 30 km. Refer to Liggio et al. (2016) for the Google Earth image that shows flight track of F_9/4 and locations of the multiple OS facilities. Altitude shown here indicates the ellipsoid height.





**Figure 2.** Mass and number size distributions of rBC for the 14 emission flights, which are derived

from SP2 measurements over the OS facilities. Results from flight tracks between the airport and





OS facilities are not involved in the analysis. Measurement date and the targeted OS facilities (1−3) are also shown for each flight. MMD, NMD, mass and number distribution widths, which are determined by lognormal fits to the measurement results, are summarized in Table S1 for these emission flights.





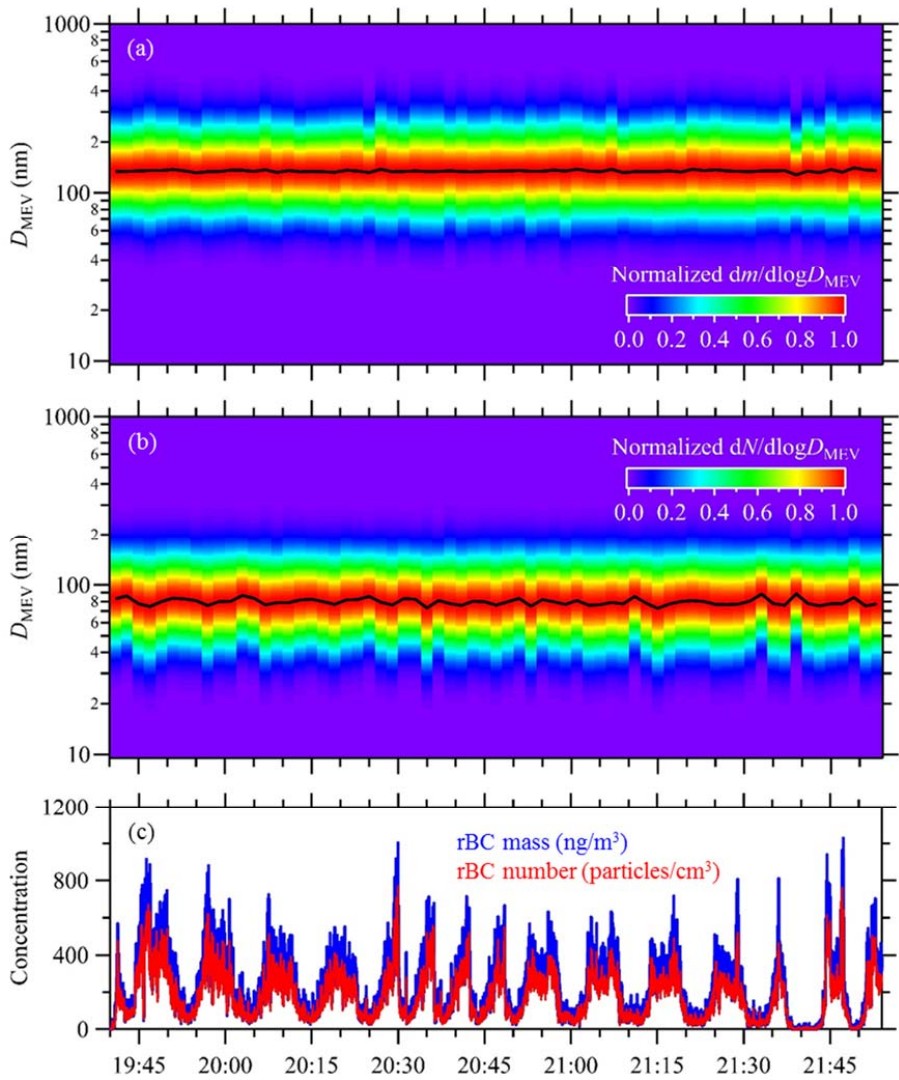

**Figure 3.** Time-resolved rBC **(a)** mass size distribution, **(b)** number size distribution, and **(c)** concentrations observed over the CNRL facility during F_8/26. Solid lines in (a) and (c) indicate MMD and NMD, respectively. The horizontal axis shows UTC time.



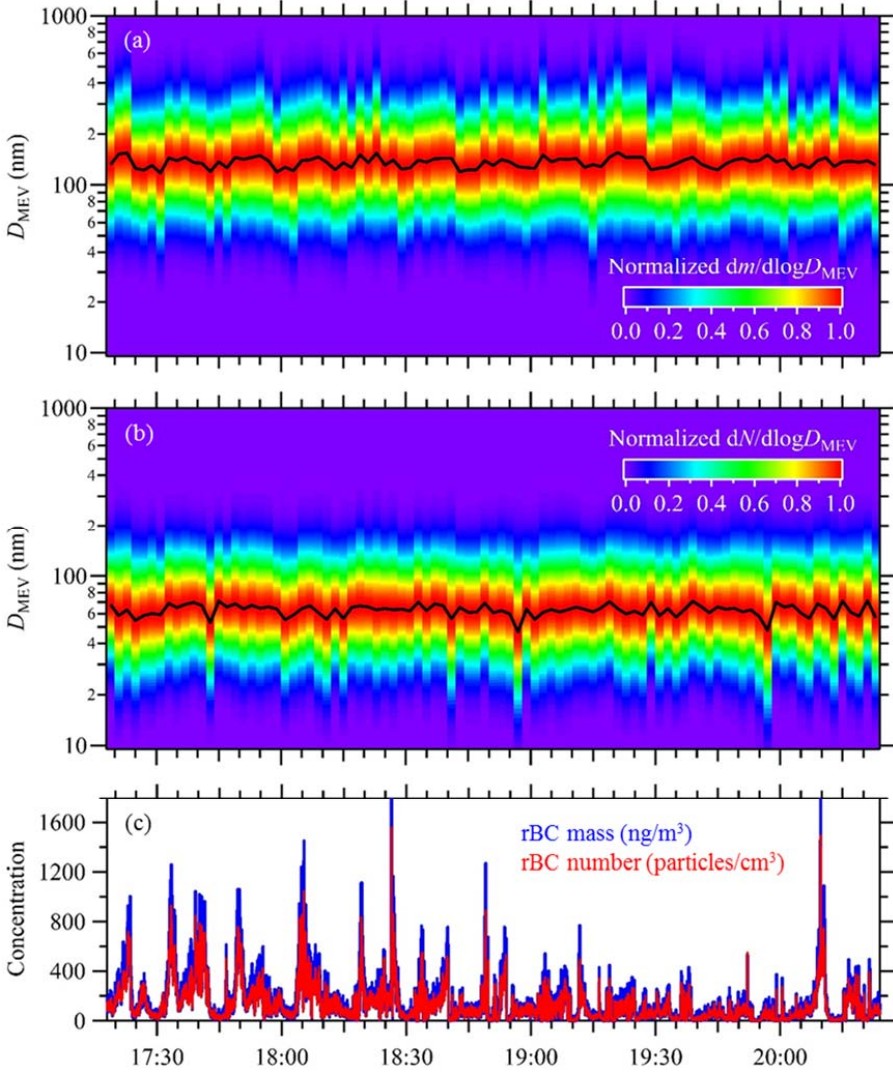

**Figure 4.** Time-resolved rBC **(a)** mass size distribution, **(b)** number size distribution, and **(c)** concentrations observed over the SUN facility during F_8/28. Solid lines in (a) and (c) indicate MMD and NMD, respectively. The horizontal axis shows UTC time. The flight track of F_8/28 is shown in Figure 1 (a).





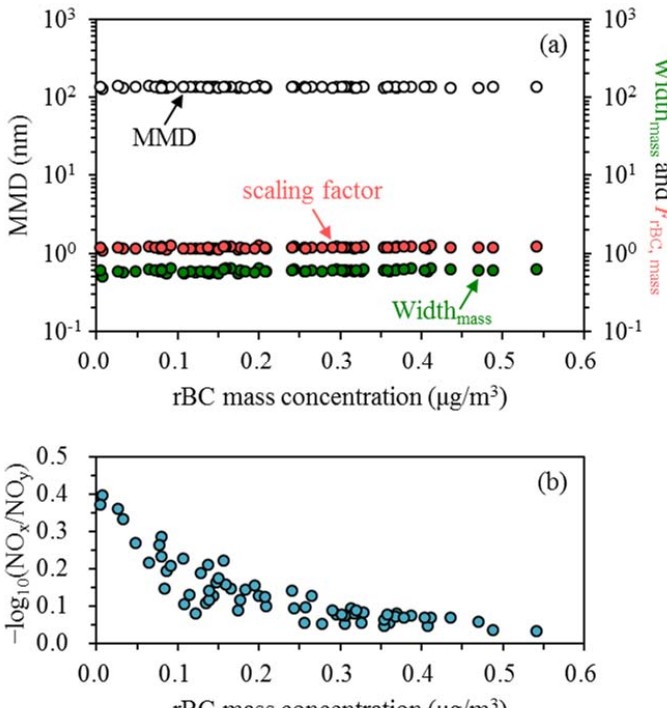

**Figure 5.** Dependences of **(a)** rBC MMD, mass distribution width (Width$_{mass}$), and mass-based scaling factor ($F_{rBC, mass}$), and **(b)** photochemical age on rBC concentration during F_8/26. Time resolution is 2 minutes for all the parameters shown here. Based on the temporal variation of 2-min averaged rBC mass concentration (Figure S1), rBC ≤ 0.1 µg/m³ can be used as an indicator for typical out-of-plume conditions during F_8/26. Results in (a) are also available in Figure S1, where rBC MMD, Width$_{mass}$ and $F_{rBC, mass}$ are presented on linear scales.



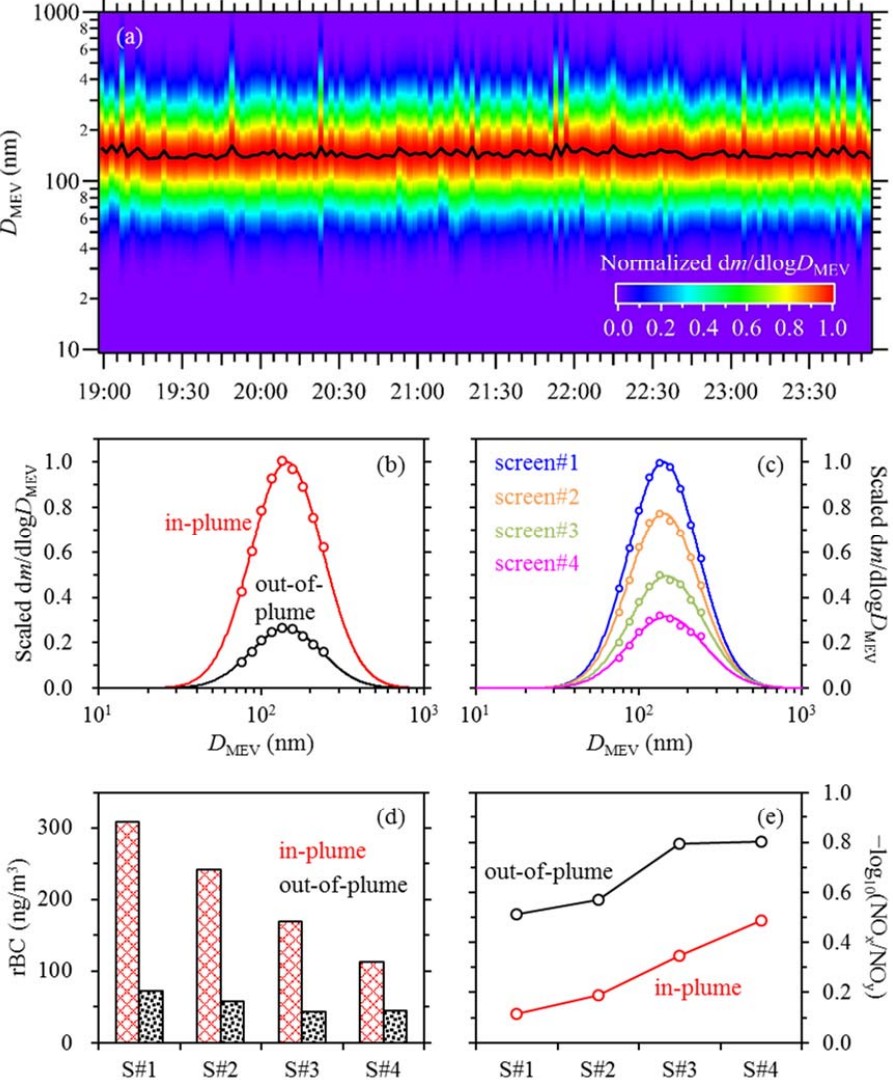

**Figure 6.** **(a)** Time-resolved rBC mass size distribution observed during the transformation flight F_9/4, **(b)** comparison of rBC mass size distribution between typical in- and out-of-plume conditions, **(c)** comparison of in-plume rBC mass size distribution among successive flight screens, and evolutions of **(d)** average rBC mass concentration and **(e)** photochemical age from screen #1 (S#1) to screen #4 (S#4). Scaling of out-of-plume rBC size distribution in (b), scaling of rBC size distributions for screens #2 to #4 in (c), and reason for excluding results from screen #5 in (c) to (e) are explained in the text.





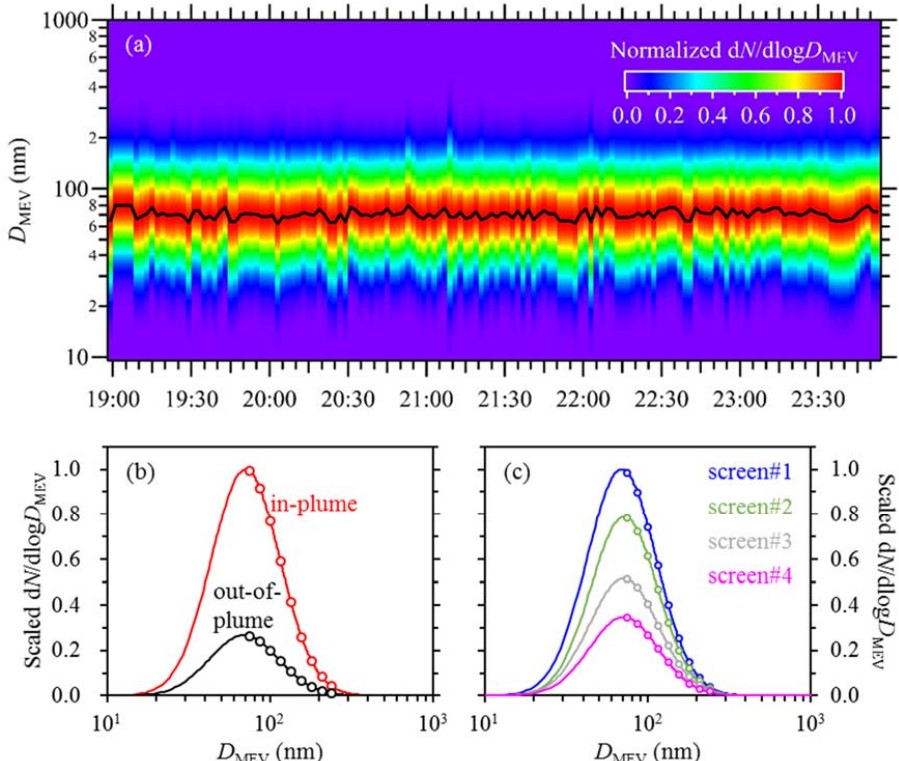

**Figure 7. (a)** Time-resolved rBC number size distribution observed during the transformation flight F_9/4, **(b)** comparison of rBC number size distribution between typical in- and out-of-plume conditions, and **(c)** comparison of in-plume rBC number size distribution among successive flight screens. Scaling of out-of-plume rBC size distribution in (b) and scaling of rBC size distributions for screens #2 to #4 in (c) are explained in the text.



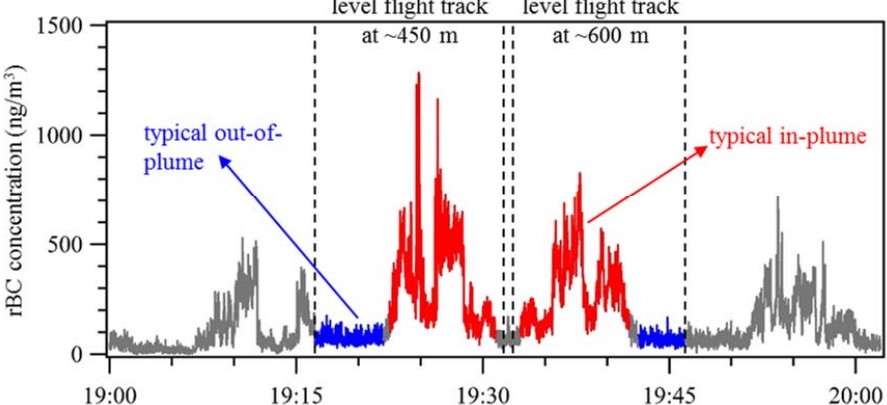

**Figure 8.** Identification of typical in- and out-of-plume conditions for two level flight tracks at ~ 450 and 600 m (in terms of ellipsoid height, equivalent to ~150 and 300 m above ground) on the first virtual screen of the transformation flight F_9/4.



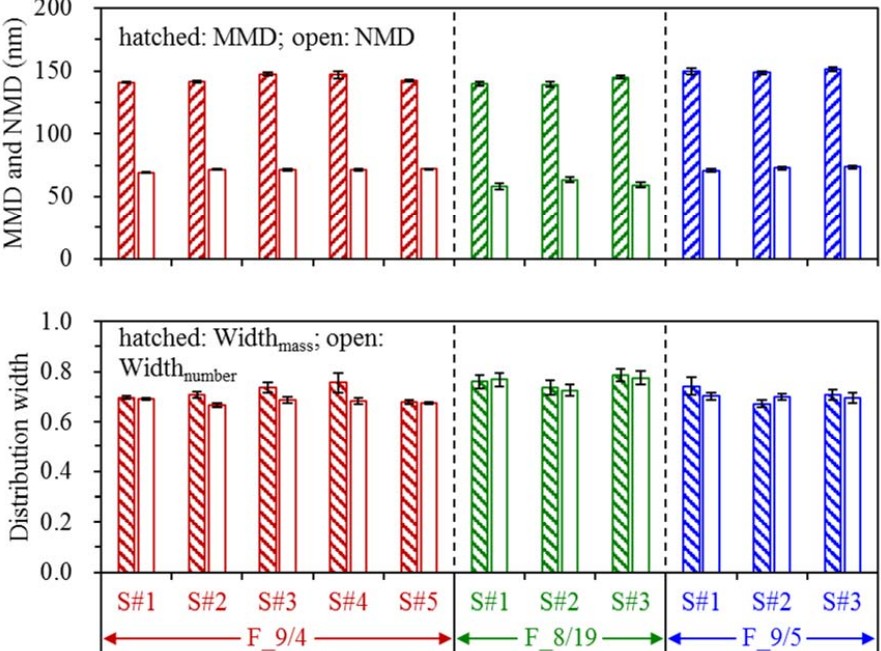

**Figure 9.** In-plume rBC MMD and NMD (upper pannel), and mass and number distribution widths (Width_mass and Width_number; lower panel) derived from successive flight screens of the three transformation flights. The results are also available in Table S2.




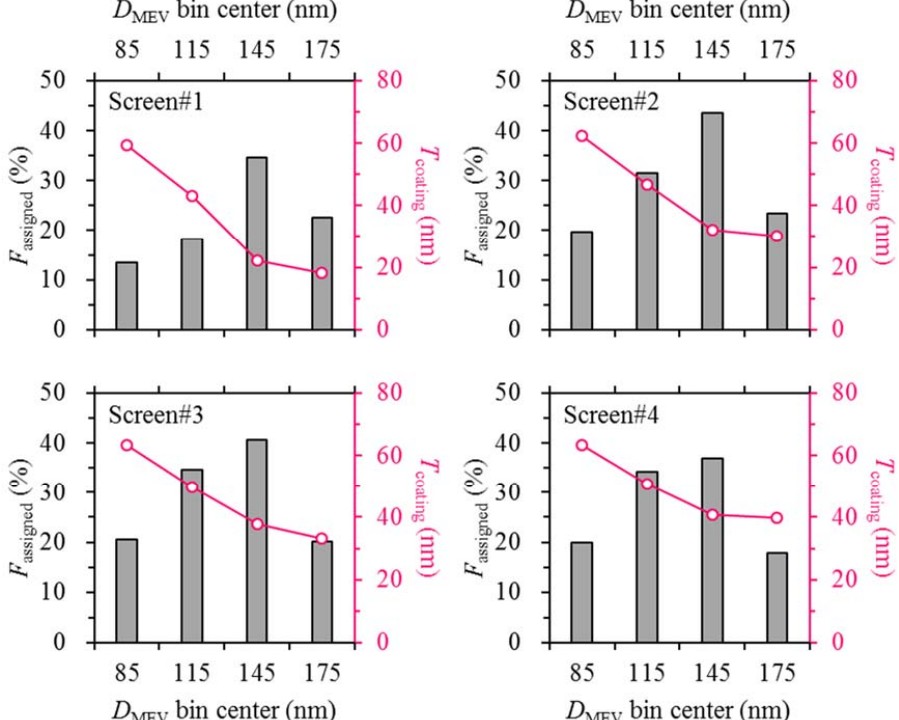

**Figure 10.** Dependence of coating thickness ($T_{coating}$) on rBC core size ($D_{MEV}$) for successive flight screens of the transformation flight F_9/4. To derive the dependence, rBC containing particles detected by the SP2 are divided into four equal-width bins according to their core sizes ($D_{MEV}$), the centers of which are 85, 115, 145, and 175 nm, respectively. The lower edge of the first $D_{MEV}$ bin is 70 nm, corresponding to the SP2's lower detection limit of $D_{MEV}$; the upper edge of the last $D_{MEV}$ bin is 190 nm. The $D_{MEV}$ range of 70 to 190 nm accounts for approximately 95% of the detected rBC cores. For each $D_{MEV}$ bin, the fraction of rBC cores that can be assigned a coating thickness ($F_{assigned}$, in %) is also shown.





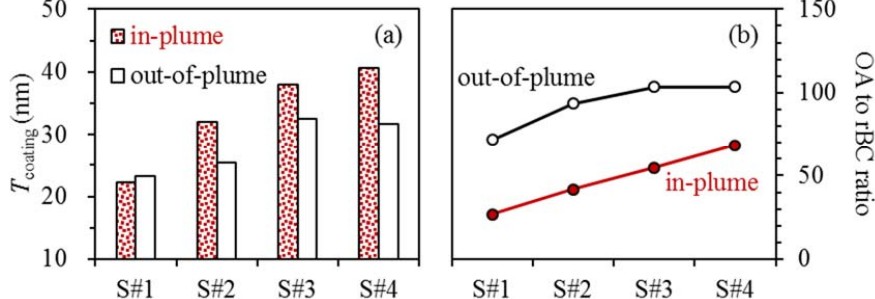

**Figure 11.** Evolutions of **(a)** coating thickness ($T_{coating}$) for rBC cores in the $D_{MEV}$ range of 130−160 nm and **(b)** OA to rBC ratio (OA/rBC) during the transformation flight F_9/4. Only medians are shown for $T_{coating}$ and OA/rBC. Quantitative discussions on OA/rBC have been presented elsewhere (Liggio et al., 2016), whereas statistical results are shown in Figure S3 for $T_{coating}$ measured in F_9/4 (together with $T_{coating}$ measured in the other two transformation flights).





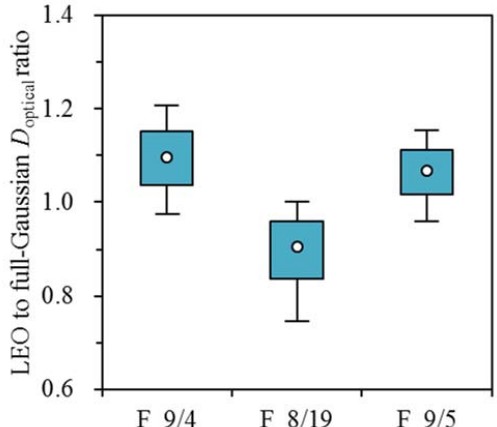

**Figure 12.** Relationships between optical sizes ($D_{optical}$) retrieved from the LEO and full-Gaussian fits for non-rBC containing particles observed during the three transformation flights.