# Peer review of "Size distribution and coating thickness of black carbon from the Canadian oil sands operations"

_Atmospheric Chemistry and Physics, 2017_

## Referee Comment (RC1) · Anonymous Referee #1 · 5 Dec 2017

In this study the authors present results from aircraft measurement on BC in the Canada oil sand region. The BC size distribution was investigated by calculating particle size from the measured mass concentration and density using SP2 instrument. The BC concentration and size distribution in and out-of-plume in the OC region and downwind area were studied and compared. The number and mass size distribution did not show significant temporal differences. Some interesting and valuable information was obtained from this BC study on the OS region. The manuscript is well-organized and clearly presented. I'd like to suggest the acceptance of this manuscript after a minor revision.

[Figure]

Line 169- is the width of "0.7" that geometric standard deviation, or coefficient of variation?

Liens 174-184, it would be better for readability and easy in a comparison if this information can be present in a table or well-designed figure.

Lines 190-191, BC mass, or number concentration distribution?

Line 203, suggest to rephrase as "including results from the present study"

Line 203, any suggestion on the variation of this 60 nm proposed?

Line 222-223, suggest to revise as "mean negligible difference in the size distribution between the in- and out-of-plume over the OS region".

Line 230-232, the information on other measurements during the flight may be necessary to be mentioned in the Method section.

Lines 300-305, source types, species present in the ambient air, and the degree of aging are all factors can significantly affect the change of BC size distribution.

Line 338, Figure S2 or Table S2?

Figure 1- is it possible to place a real map in this figure?
* * *

---

## Referee Comment (RC2) · D. Baumgardner (Referee) · 24 Dec 2017

This manuscript describes the evaluation of the physical properties of rBC particles produced by activities related to the Canadian oil fields. The analysis is detailed and the document is well written. The conclusions are supported by the observations and there are several interesting results that provide useful recommendations for the climate modeling community, as well as suggestions for quality assurance for those who do rBC measurements with the SP2.

There are several minor issues and questions that I would like addressed, as well as one fairly major omission that needs to be added.

[Figure]

Major omission In general, any study that uses measurements needs to include an error analysis that describes the limitations and uncertainties of the sensing technique. In this presentation, no mention is made of how the aerosols are sampled. I understand that this is probably already done in companion papers, but it is necessary here in order to understand any losses/enhancements that may occur due to the inlet system that is implemented. What is the probability of evaporative losses of the material that coats the rBC? Are there any bends in the sample lines where particles can be lost and what are the losses by diffusion to the walls?

Minor issues and questions

Why were there no passes made upwind of the oil sands? Although I understand that the environment outside the emissions plume is likely similar to the environment upwind of the oil sands, I am quite surprised that the flight plan designers did not consider the need for baseline data upwind that would decisively show how much the downwind aerosol and gas concentrations were elevated over the background. If I was an apologist/defender of the companies operating the oil sand project, I would be asking that question, as well.

As a suggestion, and it should be included only if it provides additional information that is not already in the paper, the authors should look at the ratios of number and derived mass between the rBC and non-rBC measured just by the SP2. This might be a useful indicator of mixing or coagulation processes with age.

Nothing is mentioned about the meteorological conditions for the days of each flight that might have changed the patterns of turbulence, mixing and removal. Were any of the legs in the mixed layer?

The comparison of the non-rBC size from light scattering using the LEO and Gaussian fit is a very good quality assurance procedure that should be followed by anyone analyzing SP2 data and deriving coating thicknesses. I think that this needs to be reiterated in the conclusions.

Line 29: "...a type of unconventional petroleum deposit". Not sure this is relevant unless this type of deposit produces more chance of elevated pollution than oter types.

Line 74: "magnitude" By mass or number concentration?

Line 129: What is the rational for restricting the Dmev range from 70-260 nm?

Line 134: What is the rational for using this scaling method and has this approach been previously used by others?

Figure 5a: Why is MMD on a log scale with 4 orders of magnitude? Wouldn't a linear scale be a better choice to see if indeed there were and shifts? Also, I think that there should be standard deviation bars with these symbols.

Line 219: Change "were" to "was"

Line 303: Scavenging of rBC by diffusion and inertia should be mentioned. The curve fitting masks any broadening that might show up because of these processes. I suggest looking at the ratios of mass of rBC> 100 nm to < 100 nm. This ratio might change with aging due to coagulation.

Line 364: is the OA to rBC ratio by number or mass?

---

## Referee Comment (RC3) · Anonymous Referee #3 · 26 Dec 2017

This paper provides a case study of size distributions and, to a lesser extent, coating thickness on refractory black carbon particles over oil sands activities in Canada. I think this work is publishable, after the following concerns are addressed.

L10: This paper does not address mixing state, so much as it addresses the properties of rBC particles. I suggest that the authors revise to make more specific for their particular study.

L21: While this is technically correct, the authors are presenting this as if it is new information. This has been known since some of the first SP2 measurements.

L23: The meaning of "consistent" is not clear. Does this mean the same shape? Same

[Figure]

MMD? Same width?

L26: The meaning of "doubled" is left ambiguous here. Did the coating thickness increase from 1 to 2 nm? Or from 50 to 100 nm? These are both doublings, but with very different implications. More detail is welcome.

L27: How can the authors be sure that the apparent increase is not due to entrainment of background air?

Abstract: I suggest the abstract is revised to make it abundantly clear that the BC is derived from activites associated with mining of the OS, and not from the OS directly. This differs from some of the other emissions that are observed in this region.

L82: I'm not sure this is necessarily the case. I don't think this, for example. The authors here set up an argument simply to shoot it down. I suggest they focus on what they observed, and leave discussion for later in the manuscript.

Fig. 2: It is unclear to me why there are so few points on these graphs. The SP2 measures at much higher size resolution than is shown here.

L115: The meaning of "virtual screen" is unclear to me. This could be clarified. Consequently, I have a difficult time following the discussion starting line 270 or so. The authors could be clearer about what they are doing.

L123: A better citation regarding calibration would probably be one of the AMT papers by Laborde. Although I suppose they don't use regal black. . .

L129: The meaning of the "conversion" is unclear to me. Do the authors mean that they measure over the per-particle mass range X-Y fg/p and this corresponds to 70-260 nm? Could be clearer.

L134: Are the authors saying that a single, campaign average for FrBC was found and applied? Or was it flight specific? Or something else?

L135: This paragraph needs to include discussion of the limitations of the scattering

method for coating thickness. The SP2 can only determine coating thicknesses for rBC particles with core diameters above some minimum size, typically around 150-160 nm. This is a limitation of the mismatch between the incandescence and scattering lower limit. This is discussed later, but it belongs in the methods. There is no reason, in my opinion, to even present Fig. 10 (since it is wrong, as the authors note later). There is too great of potential for this to be misinterpreted by people who do not read the paper carefully. This figure must be removed from the paper if it is to be published. Or, perhaps, it could be moved to the methods and discussed here as a method limitation. But it does not belong in the results and discussion.

L174: This very long sentence would be better as a table. Or could at least be supported by a table.

L185: For any study that reported a MMD and a width, the NMD can be calculated. So, even if not directly reported the information is, most likely, easily obtainable. I suggest that the authors go through the effort of extracting this information and including it as part of a table.

L190: This has been long known. The authors should rephrase to state "Our results support the well-known suggestion that rBC from fossil fuel is smaller than biomass burning" or something like that.

L202: While I generally agree with the authors, they must note here that their measurements are limited by an ability to measure below ∼70 nm. They do not know if there is some other mode at smaller sizes. Most likely there is not. But their data cannot prove this one way or another.

Fig. 5: The authors should change panel (a) to have two different axes ranges. Right now the range for the width and FrBC are way too large. This should be changed to a range of 0-1. And the MMD axis to the range 100-200.

The authors should strongly consider revising their definition of a log-normal distribution to use the more common formulation from e.g. Seinfeld and Pandis that includes a 1/sqrt(sigma) in the prefactor. This makes the widths vary from 1 to greater than 1, and is much more commonly used by e.g. climate models (which seems to be a target of this study).

L235: if the authors happen to have CO measurements from the flights then they can explicitly test this dilution hypothesis.

General: The authors are providing more sig figs than appropriate (e.g. L263). This should be revised.

Figures 9: This figure works alright, but would probably work better as set of box and whisker plots, if the authors so choose.

L299: Again, this is not generally believed. I don't believe this. I just take this to mean that background measurements are more impacted by biomass burning emissions than are urban measurements for many of the measurements that have been made. In the current study, the authors are simply sampling a particular part of the atmosphere where this is not the case. It seems that in their environment that the background rBC is dominated by, most likely, emissions from the OS activities. Thus, the size distribution of the background and plume look similar. The authors discussion here focuses much too much on "processing," in my opinion, when the bigger issue is "emissions." I think this paragraph needs revision. Overall, I think that the authors are over complicating something that seems to me quite simple.

L338: I have substantial concerns that the 130 nm particles are still too small for robust sizing that will be (mostly) bias free. Scattering by an e.g. 120 nm rBC particle with a 5 nm coating (130 nm total) will be very different than that from an 80 nm rBC particle with a 25 nm coating, for example. This can lead to biases in interpretation. The larger one goes, the less this is an issue. The most robust studies limit analysis of coating thickness to >150 nm, unless it is explicitly demonstrated that a smaller threshold is justified.

L352: The authors cannot simultaneously argue for more "aged" air outside the plume (and supported by NOx/NOy) and non-OS local emissions. Or, if they are going to do so, they need to make a more concrete and persuasive argument here in my opinion.

L356: Why would the authors not compare in/out of plume T* values based on their NOx/NOy, rather than the "screen"? A lot of this discussion would benefit from a more direct link to photochemical age. Right now, the concept of photochemical age is seemingly left behind to earlier discussions, but it also belongs here in my opinion.

L376: In this discussion, the authors should note more explicitly that they are working from the small-size side of things, compared to these other studies. Generally, one might expect LEO to be more robust for larger particles, with larger signal. Of course, there is a limitation because as particles get too large the scattering detector is saturated and a full Gaussian cannot be fit. Overall, the point that there is uncertainty on the order of 10% in the ooptical diameter from the LEO fit is worth reporting, but the authors should note the issue that they are using small particles.

Fig. 11b/L363: While true, the authors neglect that the OA/rBC is smaller for the in-plume conditions overall. While not all OA will be coated on rBC, that the OA/rBC is so much smaller in plume might lead one to think that the coating amount on the rBC in plume should be smaller compared to out of plume, opposite what is reported. This should be discussed, in my opinion.

---

## Author Comment (AC1) · 25 Jan 2018

We appreciate the careful consideration of our manuscript by this reviewer. We have considered the points raised and revised our manuscript accordingly. Our detailed responses and all changes that have been made are presented below.

**General comments**

In this study the authors present results from aircraft measurement on BC in the Canada oil sand region. The BC size distribution was investigated by calculating particle size from the measured mass concentration and density using SP2 instrument. The BC concentration and size distribution in and out-of-plume in the OC region and downwind area were studied and compared. The number and mass size distribution did not show significant temporal differences. Some interesting and valuable information was obtained from this BC study on the OS region. The manuscript is well-organized and clearly presented. I'd like to suggest the acceptance of this manuscript after a minor revision.

**Specific comments**

**(1)** Line 169. Is the width of "0.7" that geometric standard deviation, or coefficient of variation?

**Our response:** The fitting parameters "mass distribution width" and "number distribution width" are defined by Equations (1) and (2), respectively. Briefly, the measured masses of the individual rBC cores were first grouped into different size bins and then fitted by a lognormal curve:

$$\frac{\mathrm{d}m}{\mathrm{d}\log D_{\mathrm{MEV}}} = A_{\mathrm{mass}} \times \exp\left\{0 - \left[\frac{\ln\left(D_{\mathrm{MEV}}/X_{1,\,\mathrm{mass}}\right)}{X_{2,\,\mathrm{mass}}}\right]^2\right\} \tag{1}$$

where the fitting parameter $X_{1,\,\mathrm{mass}}$ is termed the mass median diameter (MMD), and the fitting parameter $X_{2,\,\mathrm{mass}}$ is referred to as the mass distribution width (Width$_{\mathrm{mass}}$). Similarly, rBC number-size distribution was expressed as:

$$\frac{\mathrm{d}N}{\mathrm{d}\log D_{\mathrm{MEV}}} = A_{\mathrm{number}} \times \exp\left\{0 - \left[\frac{\ln\left(D_{\mathrm{MEV}}/X_{1,\,\mathrm{number}}\right)}{X_{2,\,\mathrm{number}}}\right]^2\right\} \tag{2}$$

where the fitting parameter $X_{1,\,\mathrm{number}}$ is termed the number median diameter (NMD), and the fitting parameter $X_{2,\,\mathrm{number}}$ is referred to as the number distribution width (Width$_{\mathrm{number}}$).

In addition, for a lognormal rBC mass-size distribution, the mass distribution width (Width$_{\mathrm{mass}}$) determined by Equation (1) can be converted to the standard deviation of the

distribution ($\sigma_{\text{mass}}$) by $\sigma_{\text{mass}} = \exp\left(\text{Width}_{\text{mass}}\big/\sqrt{2}\right)$. Similarly, $\text{Width}_{\text{number}}$ can be converted to the standard deviation of a lognormal rBC number size distribution ($\sigma_{\text{number}}$) by $\sigma_{\text{number}} = \exp\left(\text{Width}_{\text{number}}\big/\sqrt{2}\right)$. For a lognormal distribution, therefore, a distribution width of ~ 0.7 corresponds to a standard deviation of ~ 1.6. The "$\text{Width}_{\text{mass}}$ vs. $\sigma_{\text{mass}}$" and "$\text{Width}_{\text{number}}$ vs. $\sigma_{\text{number}}$" relationships were added to the manuscript.

**(2)** Lines 174-184. It would be better for readability and easy in a comparison if this information can be present in a table or well-designed figure.

**Our response:** A table was added as suggested. In addition to the results from fresh urban emissions discussed in this paragraph, rBC size distributions observed for biomass burning plumes and in remote areas were also involved in this table.

**(3)** Lines 190-191. BC mass, or number concentration distribution?

**Our response:** The statement that "rBC cores emitted from fossil fuel combustion were smaller in size compared to those from biomass burning" is valid for both rBC mass and number size distributions. Detailed MMD and NMD were presented in Table 1 and discussed in the sentence following this statement.

**(4)** Line 203. Suggest to rephrase as "including results from the present study".

**Our response:** The change was made as suggested.

**(5)** Line 203. Any suggestion on the variation of this 60 nm proposed?

**Our response:** We noticed that three values (i.e., 30, 40 and 60 nm) are being used in aerosol-climate models as the NMD of black carbon emitted by fossil fuel combustion. But we were unable to estimate the uncertainties of these NMD settings, including that we proposed (i.e., 60 nm) based on the SP2 measurement results on rBC.

**(6)** Line 222-223. Suggest to revise as "mean negligible difference in the size distribution between the in- and out-of-plume over the OS region".

**Our response:** The change was made as suggested.

**(7)** Line 230-232. The information on other measurements during the flight may be necessary to be mentioned in the Method section.

**Our response:** A new section entitled "Additional dataset used" was added to the Method section, in which the measurements of $NO_x$, $NO_y$ and organic aerosol (OA) were introduced briefly. Accordingly, descriptions of $NO_x$ and $NO_y$ measurements were removed from this

paragraph.

**(8)** Lines 300-305. Source types, species present in the ambient air, and the degree of aging are all factors can significantly affect the change of BC size distribution.

**Our response:** We agree with the reviewer that in addition to the factor we mentioned, there could be other factors that can change rBC size distribution during aging. The sentence was revised to "……influences of aging on rBC size distribution may partially depend on the presence of atmospheric processes that can lead to increased rBC core mass and size in a single particle (e.g., rBC coagulation and evaporation of cloud droplets containing multiple rBC particles).".

**(9)** Line 338. Figure S2 or Table S2?

**Our response:** It should be Figure S2. This point was clarified in the revised manuscript.

**(10)** Figure 1. Is it possible to place a real map in this figure?

**Our response:** Google Earth images showing flight tracks were provided as suggested. Caption of Figure 1 was updated accordingly.

---

## Author Comment (AC2) · 25 Jan 2018

We appreciate the careful consideration of our manuscript by this reviewer. We have considered the points raised and revised our manuscript accordingly. Our detailed responses and all changes that have been made are presented below.

**General comments**

This manuscript describes the evaluation of the physical properties of rBC particles produced by activities related to the Canadian oil fields. The analysis is detailed and the document is well written. The conclusions are supported by the observations and there are several interesting results that provide useful recommendations for the climate modeling community, as well as suggestions for quality assurance for those who do rBC measurements with the SP2. There are several minor issues and questions that I would like addressed, as well as one fairly major omission that needs to be added.

**Major omission.**

In general, any study that uses measurements needs to include an error analysis that describes the limitations and uncertainties of the sensing technique. In this presentation, no mention is made of how the aerosols are sampled. I understand that this is probably already done in companion papers, but it is necessary here in order to understand any losses/enhancements that may occur due to the inlet system that is implemented. What is the probability of evaporative losses of the material that coats the rBC? Are there any bends in the sample lines where particles can be lost and what are the losses by diffusion to the walls?

**Our response:**

**(1) What is the probability of evaporative losses of the material that coats the rBC?**

While it is possible that evaporation loss of the coating cannot be ruled out during the transit from the tip of the inlet to the detection point in the SP2 instrument, we have evidence to show that this loss is not important. As shown in the figure below, the measurement of particle size distribution at the ambient condition using the wing-mounted inlet-less FSSP 300 (with a detection range of 0.3 to 20 μm in terms of particle diameter) compared well with the inboard measurement using the UHSAS instrument (with a detection range of 0.06 to 1.0 μm in terms of particle diameter) in the overlapping size range (i.e., 0.3 to 1.0 μm). The UHSAS and the SP2 shared the same aerosol inlet and sampling line. Based on this comparison, we concluded that the evaporation loss in the aerosol inlet and sampling line was negligible.

**(2) Are there any bends in the sample lines where particles can be lost and what are the losses by diffusion to the walls?**

Aerosols were sampled through an isokinetic, shrouded solid diffuser inlet (Droplet Measurement Technologies Inc., Boulder, CO, USA) with a NASA design as described in Huebert et al. (2004). All aerosol instruments inboard the aircraft shared the same inlet and sampling line. There were gentle bends in the sampling line inside the aircraft; however, the flow in the sampling line was laminar and hence loss of the particles to the wall was minimal. This conclusion is also supported by the comparison between the results from the FSSP and UHSAS.

The discussions above were reflected in the revised manuscript.

[Figure]

**Figure R2_1.** Comparison of particle number size distributions derived from wing-mounted FSSP and inboard UHSAS based on results from segments of the transformation flight conducted on August 19, 2013. These segments were flown at different downwind distances from the oil sands source area, along level flight tracks at multiple altitudes. FSSP and UHSAS measure optical sizes for particles in the diameter ranges of 0.3 to 20 μm and 0.06 to 1.0 μm, respectively, with an overlapping detection range of 0.3 to 1.0 μm. Measurement results from FSSP are shown only for the overlapping size range. For UHSAS, the measurement results exhibited a bimodal distribution for the flight segments investigated; only the mode at relatively large sizes, which is characteristic of the accumulation mode, is shown for comparison with FSSP. Although FSSP and UHSAS can be compared only at the trailing edge of the UHSAS size distribution, an agreement is observed between these two instruments in terms of both particle numbers and their size distributions. This agreement suggests that in the aerosol inlet and sampling line, both the particle loss and the evaporation loss from the particles should be minimal for the < 1.0 μm size range.

**Minor issues and questions**

**(1)** Why were there no passes made upwind of the oil sands? Although I understand that the environment outside the emissions plume is likely similar to the environment upwind of the oil sands, I am quite surprised that the flight plan designers did not consider the need for baseline data upwind that would decisively show how much the downwind aerosol and gas

concentrations were elevated over the background. If I was an apologist/defender of the companies operating the oil sand project, I would be asking that question, as well.

**Our response:** As pointed out by the reviewer, we assumed that the environment outside the oil sands plume was similar to the environment upwind of the source area. We did not conduct comprehensive measurements upwind of the oil sands during the three transformation flights, to ensure that at least three screens downwind of the source area could be achieved for each flight. Different screens indicated different transport times, different photochemical ages, and different oxidation states for the oil sands emissions. We thought that at least three different screens were required for each flight to illustrate the dependences of aerosol physical and chemical properties on photochemical age. Nonetheless, we will try to follow the reviewer's suggestion (by involving upwind screens) in the coming aircraft campaign that will be conducted over the same oil sands region.

**(2)** As a suggestion, and it should be included only if it provides additional information that is not already in the paper, the authors should look at the ratios of number and derived mass between the rBC and non-rBC measured just by the SP2. This might be a useful indicator of mixing or coagulation processes with age.

**Our response:** Mass and number concentrations of non-rBC containing particles were derived from the SP2. It was found that the ratios of non-rBC containing particles to rBC cores (based on either mass or number concentrations) indeed changed as the oil sands plumes were transported downwind. The detailed results will be presented elsewhere.

**(3)** Nothing is mentioned about the meteorological conditions for the days of each flight that might have changed the patterns of turbulence, mixing and removal. Were any of the legs in the mixed layer?

**Our response:** We agree with the reviewer that meteorological conditions could influence the patterns of turbulence, mixing and removal. As shown in Figure 3 (for the 14 emission flights) and Figure 10 (for the 3 transformation flights) in the revised manuscript, rBC size distributions were in general consistent among different measurement dates that had different meteorological conditions, indicating negligible influence of meteorological parameters on rBC size distribution during the campaign. Therefore, we prefer not to discuss the variations of meteorological conditions, since they could not be linked to the observational results presented in this manuscript. However, it should be noted that the measurements conducted during the aircraft campaign were not influenced by wet removal processes such as precipitation, which

might be partially responsible for the lack of daily variation in rBC size distribution. This point was clarified in the revised manuscript.

In addition, the measurements were primarily conducted in the mixed layer, whereas some level flight tracks reached above the mixed layer (Li et al., 2017). The point was also clarified in the revised manuscript.

**(4)** The comparison of the non-rBC size from light scattering using the LEO and Gaussian fit is a very good quality assurance procedure that should be followed by anyone analyzing SP2 data and deriving coating thicknesses. I think that this needs to be reiterated in the conclusions.

**Our response:** This point was reiterated in the Conclusions section as suggested.

**(5)** Line 29. "...a type of unconventional petroleum deposit". Not sure this is relevant unless this type of deposit produces more chance of elevated pollution than other types.

**Our response:** This should be relevant because recovery techniques used in the oil sands industry (e.g., surface mining) are different from those used for conventional petroleum deposit.

**(6)** Line 74. "magnitude" By mass or number concentration?

**Our response:** "Magnitude" mentioned here was in terms of mass. This point was clarified in the revised manuscript.

**(7)** Line 129. What is the rational for restricting the $D_{MEV}$ range from 70-260 nm?

**Our response:** This $D_{MEV}$ range corresponded to the SP2's detection range of single particle rBC core mass (i.e., ~ 0.3−16 fg), under the commonly-used assumption that the density of the rBC core is 1.8 g/cm$^3$. This sentence was revised to "Based on this $\rho$ value, the SP2's detection range for single particle rBC core mass (~ 0.3−16 fg) corresponded to an rBC size detection range of ~ 70−260 nm in terms of $D_{MEV}$.", which should be clearer.

**(8)** Line 134. What is the rational for using this scaling method and has this approach been previously used by others?

**Our response:** This scaling method has been commonly used in the previous SP2-based studies, which was initially introduced by Schwarz et al. (2006). This reference was cited when describing the scaling method.

**(9)** Figure 5a. Why is MMD on a log scale with 4 orders of magnitude? Wouldn't a linear scale be a better choice to see if indeed there were and shifts? Also, I think that there should be standard deviation bars with these symbols.

**Our response:** This figure was revised to show MMD as well as the other two parameters on linear scales. In addition, standard deviations were fairly small for the time-resolved MMD and mass distribution width, within ± 5 nm and ± 0.06 respectively. Therefore, we prefer to describe the standard deviations in the figure caption, rather than showing them in the figure.

**(10)** Line 219. Change "were" to "was".

**Our response:** The change was made as suggested.

**(11)** Line 303. Scavenging of rBC by diffusion and inertia should be mentioned. The curve fitting masks any broadening that might show up because of these processes. I suggest looking at the ratios of mass of rBC> 100 nm to < 100 nm. This ratio might change with aging due to coagulation.

**Our response:** We agree with the reviewer that in addition to the factor we mentioned (i.e., evaporation of cloud droplets containing multiple rBC particles), there could be other factors (e.g., rBC coagulation) that can change rBC size distribution during aging. Following the reviewer's suggestion, the ratios of rBC cores with $D_{MEV}$ above 100 nm to those with $D_{MEV}$ below 100 nm were calculated based on number concentrations. As shown in the figure below, this ratio was fairly constant as the oil sands plume was transported downwind, indicating that rBC coagulation should be insignificant during the plume aging. This is not surprising, given that plume aging was accompanied by dilution. Based on the discussions above, the related descriptions were revised to "……influences of aging on rBC size distribution may partially depend on the presence of atmospheric processes that can lead to increased rBC core mass and size in a single particle (e.g., rBC coagulation and evaporation of cloud droplets containing multiple rBC particles). In this study, it appears that no such processes were at play……".

[Figure]

**Figure R2_2.** Ratios of rBC cores with $D_{MEV}$ between 100 and 260 nm to those with a $D_{MEV}$ range of 70−100 nm based on number concentrations. S#1 to S#5 indicate successive flight screens of the transformation flight F_9/4, with increasing downwind distances from the OS source area. Variation of this ratio is within 2.5%, indicating insignificant rBC coagulation during the plume aging.

**(12)** Line 364. Is the OA to rBC ratio by number or mass?

**Our response:** The ratio was based on mass concentrations. This point was clarified in the revised manuscript.

**References**

Huebert, B. J., Howell, S. G., Covert, D., Bertram, T., Clarke, A., Anderson, J. R., Lafleur, B. G., Seebaugh, W. R., Wilson, J. C., Gesler, D., Blomquist, B., and Fox, J.: PELTl: measuring the passing efficiency of an airborne low turbulence aerosol inlet, Aerosol Sci. Technol., 38, 803–826, 2004.

Li, S. M., Leithead, A., Moussa, S. G., Liggio, J., Moran, M. D., Wang, D., Hayden, K., Darlington, A., Gordon, M, Staebler, R., Makar, P. A., Stroud, C. A., McLaren, R., Liu, P., O'Brien, J., Mittermeier, R. L., Zhang, J., Marson, G., Cober, S. G., Wolde, M., and Wentzell, J.: Differences between measured and reported volatile organic compound emissions from oil sands facilities in Alberta, Canada, Proc. Natl. Acad. Sci. U.S.A., 114, E3756–E3765, 2017.

Schwarz, J. P., Gao, R. S., Fahey, D. W., Thomson, D. S., Watts, L. A., Wilson, J. C., Reeves, J. M., Darbeheshti, M., Baumgardner, D. G., Kok, G. L., Chung, S. H., Schulz, M., Hendricks, J., Lauer, A., Karcher, B., Slowik, J. G., Rosenlof, K. H., Thompson, T. L., Langford, A. O., Loewenstein, M., and Aikin, K. C.: Single-particle measurements of midlatitude black carbon and light-scattering aerosols from the boundary layer to the lower stratosphere, J. Geophys. Res., 111, D16207, doi:10.1029/2006JD007076, 2006.

---

## Author Comment (AC3) · 25 Jan 2018

We appreciate the careful consideration of our manuscript by this reviewer. We have considered the points raised and revised our manuscript accordingly. Our detailed responses and all changes that have been made are presented below.

**General comments**

This paper provides a case study of size distributions and, to a lesser extent, coating thickness on refractory black carbon particles over oil sands activities in Canada. I think this work is publishable, after the following concerns are addressed.

**Specific comments**

**(1)** L10. This paper does not address mixing state, so much as it addresses the properties of rBC particles. I suggest that the authors revise to make more specific for their particular study.

**Our response:** We agree with the reviewer that this manuscript did not address mixing state of particles and instead, only the mixing state of rBC containing particles was investigated (based on coating thickness). It was confirmed that coating thickness retrieved from SP2 has been commonly used as a measure of mixing state for black carbon aerosols (e.g., Schwarz et al., 2008; Kondo et al., 2011; Laborde et al., 2013). It was also confirmed that the term "mixing state" was always used for rBC containing particles or black carbon aerosols throughout the manuscript. Therefore, we think the use of "mixing state" in this manuscript will not cause misunderstandings.

**(2)** L21. While this is technically correct, the authors are presenting this as if it is new information. This has been known since some of the first SP2 measurements.

**Our response:** We agree with the reviewer that this manuscript was not the first one challenging the NMD settings for fossil fuel BC in aerosol-climate models. But we think this manuscript indeed provides additional information for this argument, based on a distinct type of BC source (i.e., oil sands operations) that has rarely been investigated. Nonetheless, we removed the related descriptions, which appear to imply that this point is a new finding of the present study, from the Conclusions section.

**(3)** L23. The meaning of "consistent" is not clear. Does this mean the same shape? Same MMD? Same width?

**Our response:** "Consistent" means the same MMD, NMD and the corresponding distribution widths. This point was clarified in the revised manuscript.

**(4)** L26. The meaning of "doubled" is left ambiguous here. Did the coating thickness increase from 1 to 2 nm? Or from 50 to 100 nm? These are both doublings, but with very different implications. More detail is welcome.

**Our response:** The coating thickness increased from ~ 20 to 40 nm within three hours when the oil sands plume was transported over a distance of 90 km form the source area. Coating thickness values were added as suggested.

**(5)** L27. How can the authors be sure that the apparent increase is not due to entrainment of background air?

**Our response:** As shown in Figure 12a and discussed in Section 3.4, coating thicknesses were in general smaller for rBC particles in the background air compared to the in-plume rBC cores. Therefore, the increase of coating thickness for in-plume rBC cores observed during aging could not be attributed to the entrainment of background air.

**(6)** Abstract. I suggest the abstract is revised to make it abundantly clear that the BC is derived from activities associated with mining of the OS, and not from the OS directly. This differs from some of the other emissions that are observed in this region.

**Our response:** The change was made as suggested: "we focused on BC emissions from the oil sands (OS) surface mining activities in northern Alberta".

**(7)** L82. I'm not sure this is necessarily the case. I don't think this, for example. The authors here set up an argument simply to shoot it down. I suggest they focus on what they observed, and leave discussion for later in the manuscript.

**Our response:** This sentence was removed as suggested.

**(8)** Fig. 2. It is unclear to me why there are so few points on these graphs. The SP2 measures at much higher size resolution than is shown here.

**Our response:** In this study, individual rBC cores detected during a specific period were first grouped into different size bins and then fitted by a lognormal curve. The same size bin was used throughout the present study, e.g., for the facility-integrated results shown in Figure 3 (i.e., Figure 2 in the original manuscript) and for the time-resolved results shown in Figures 4, 5, 7, and 8. This point was clarified in the revised manuscript.

**(9)** L115. The meaning of "virtual screen" is unclear to me. This could be clarified. Consequently, I have a difficult time following the discussion starting line 270 or so. The

authors could be clearer about what they are doing.

**Our response:** A virtual screen corresponds to a specific downwind distance from the oil sands source area and consists of level flight tracks perpendicular to the wind direction at multiple altitudes. This sentence was added to the caption of Figure 1 which shows an example for the transformation flights involving multiple virtual screens. Moreover, a composite Google Earth image (as shown below) was provided for an easier understanding of transformation flight and virtual screens.

[Figure]

**Figure R3_1.** Composite Google Earth image showing flight track (colored by rBC mass concentration) for the transformation flight F_9/4.

**(10)** L123. A better citation regarding calibration would probably be one of the AMT papers by Laborde. Although I suppose they don't use regal black.

**Our response:** We agree with the reviewer that Laborde et al. (2012) was highly relevant to SP2 calibration. But we prefer to cite Cappa et al. (2012) here because regal black was used in it to calibrate the SP2.

**(11)** L129. The meaning of the "conversion" is unclear to me. Do the authors mean that they

measure over the per-particle mass range X-Y fg/p and this corresponds to 70-260 nm? Could be clearer.

**Our response:** This sentence was revised to : "……the SP2's detection range for single particle rBC core mass ($\sim 0.3-16$ fg) corresponded to an rBC size detection range of $\sim 70-260$ nm in terms of $D_{MEV}$.", which should be clearer.

**(12)** L134. Are the authors saying that a single, campaign average for $F_{rBC}$ was found and applied? Or was it flight specific? Or something else?

**Our response:** Flight specific $F_{rBC}$ were used to scale rBC concentrations in this study. This point was clarified in the revised manuscript.

**(13)** L135. This paragraph needs to include discussion of the limitations of the scattering method for coating thickness. The SP2 can only determine coating thicknesses for rBC particles with core diameters above some minimum size, typically around 150-160 nm. This is a limitation of the mismatch between the incandescence and scattering lower limit. This is discussed later, but it belongs in the methods. There is no reason, in my opinion, to even present Fig. 10 (since it is wrong, as the authors note later). There is too great of potential for this to be misinterpreted by people who do not read the paper carefully. This figure must be removed from the paper if it is to be published. Or, perhaps, it could be moved to the methods and discussed here as a method limitation. But it does not belong in the results and discussion.

**Our response:** Discussions on the limitations of the scattering method for coating thickness determination (including the corresponding figure), which were presented in the Results and Discussion section in the original manuscript, were moved to the Method section as suggested.

**(14)** L174. This very long sentence would be better as a table. Or could at least be supported by a table.

**Our response:** A table was added as suggested, which also included rBC NMD following the suggestion given in Comment# 15.

**(15)** L185. For any study that reported a MMD and a width, the NMD can be calculated. So, even if not directly reported the information is, most likely, easily obtainable. I suggest that the authors go through the effort of extracting this information and including it as part of a table.

**Our response:** rBC NMD were calculated as suggested for the studies which reported both the rBC MMD and mass distribution width. The calculated results were summarized in Table 1, together with reported NMD values. The NMD range representative for urban emissions was

updated accordingly.

**(16)** L190. This has been long known. The authors should rephrase to state "Our results support the well-known suggestion that rBC from fossil fuel is smaller than biomass burning" or something like that.

**Our response:** The change was made as suggested.

**(17)** L202. While I generally agree with the authors, they must note here that their measurements are limited by an ability to measure below ~ 70 nm. They do not know if there is some other mode at smaller sizes. Most likely there is not. But their data cannot prove this one way or another.

**Our response:** We agree with the reviewer that the SP2 used in this study could not detect rBC cores below ~ 70 nm and thus, we have to assume that there was only one size distribution mode for the rBC cores, which might be invalid. This limitation was discussed in the revised manuscript: "However, there is also a need to evaluate the unimodal assumption for black carbon size distribution (Liggio et al., 2012; Buffaloe et al., 2014), given the SP2's limited detection range of rBC core size.".

**(18)** Fig. 5. The authors should change panel (a) to have two different axes ranges. Right now the range for the width and $F_{rBC}$ are way too large. This should be changed to a range of 0-1. And the MMD axis to the range 100-200.

**Our response:** The figure was revised as suggested. The axis ranges we finally chose were slightly different from the suggested values, to more clearly distinguish different data series in the figure.

**(19)** The authors should strongly consider revising their definition of a log-normal distribution to use the more common formulation from e.g. Seinfeld and Pandis that includes a 1/sqrt(sigma) in the prefactor. This makes the widths vary from 1 to greater than 1, and is much more commonly used by e.g. climate models (which seems to be a target of this study).

**Our response:** We agree with the reviewer that different fitting parameters are being used to describe a lognormal distribution. As pointed out by the reviewer, there are two fitting parameters that can be converted between each other, i.e., the distribution width used in this manuscript as well as in some other SP2-based studies (e.g., McMeeking et al., 2010) and the standard deviation used by Seinfeld and Pandis in their book entitled "Atmospheric Chemistry and Physics". For a lognormal rBC mass-size distribution, the mass distribution width

(Width$_{mass}$) can be converted to the standard deviation of the distribution ($\sigma_{mass}$) by $\sigma_{mass} = \exp\left(\text{Width}_{mass}/\sqrt{2}\right)$. Similarly, Width$_{number}$ can be converted to the standard deviation of a lognormal rBC number size distribution ($\sigma_{number}$) by $\sigma_{number} = \exp\left(\text{Width}_{number}/\sqrt{2}\right)$. For a lognormal distribution, therefore, a distribution width of ~ 0.7 corresponds to a standard deviation of ~ 1.6. The "Width$_{mass}$ vs. $\sigma_{mass}$" and "Width$_{number}$ vs. $\sigma_{number}$" relationships were added to the manuscript.

**(20)** L235. If the authors happen to have CO measurements from the flights then they can explicitly test this dilution hypothesis.

**Our response:** We agree with the reviewer that CO data, if available, can help to evaluate the dilution hypothesis. This dilution hypothesis was removed to avoid any misunderstanding.

**(21)** General. The authors are providing more sig figs than appropriate (e.g. L263). This should be revised.

**Our response:** The change was made as suggested.

**(22)** Figures 9. This figure works alright, but would probably work better as set of box and whisker plots, if the authors so choose.

**Our response:** We prefer to keep this figure as is. First, time-resolved lognormal fitting was not performed for all the flights. Second, the fitting parameters shown in this figure, e.g., for F_9/4, correspond to the various lognormal curves shown in Figures 7 (c) and 8 (c); and these fitting curves represent screen-averaged rather than time-resolved results.

**(23)** L299. Again, this is not generally believed. I don't believe this. I just take this to mean that background measurements are more impacted by biomass burning emissions than are urban measurements for many of the measurements that have been made. In the current study, the authors are simply sampling a particular part of the atmosphere where this is not the case. It seems that in their environment that the background rBC is dominated by, most likely, emissions from the OS activities. Thus, the size distribution of the background and plume look similar. The authors' discussion here focuses much too much on "processing," in my opinion, when the bigger issue is "emissions." I think this paragraph needs revision. Overall, I think that the authors are over complicating something that seems to me quite simple.

**Our response:** We agree with the reviewer that the statement "It is commonly believed that…… " is exaggerated. We also agree with the reviewer that the in- vs. out-of-plume comparison of rBC size distributions was less straightforward to support our argument that "not

all aging processes will change rBC size distribution". This paragraph was revised to: "……the rBC MMD was found to be 20 nm higher for aged urban plumes from Nagoya, Japan compared to fresh emissions from the same urban area (Moteki et al., 2007). Therefore, it has been argued that rBC size distribution tends to shift toward larger sizes during aging (e.g., McMeeking et al., 2010). Results from the present study, especially the comparison of rBC size distributions among successive flight screens (Figure 10), indicate that this is not necessarily the case……."

**(24)** L338. I have substantial concerns that the 130 nm particles are still too small for robust sizing that will be (mostly) bias free. Scattering by an e.g. 120 nm rBC particle with a 5 nm coating (130 nm total) will be very different than that from an 80 nm rBC particle with a 25 nm coating, for example. This can lead to biases in interpretation. The larger one goes, the less this is an issue. The most robust studies limit analysis of coating thickness to >150 nm, unless it is explicitly demonstrated that a smaller threshold is.

**Our response:**

[Figure]

**Figure R3_2.** Comparison of coating thicknesses ($T_{coating}$) for in-plume rBC cores in two different $D_{MEV}$ ranges (i.e., 130−160 nm and 160−190 nm) during the transformation flight F_9/4. Results from successive flight screens are shown separately. Coating thicknesses determined for 160−190 nm rBC cores are smaller than $T_{coating}$ of 130−160 nm rBC cores, which can be attributed to the limitation that the detection range of $T_{coating}$ is rBC $D_{MEV}$ dependent. Evolutions of $T_{coating}$ exhibit the same pattern for rBC cores in the two different $D_{MEV}$ ranges. However, both the count of rBC cores that can be assigned a coating thickness and the fraction of rBC cores that can be assigned a coating thickness are higher for the $D_{MEV}$ range of 130−160 nm. Therefore, 130−160 nm rBC cores are used in the main manuscript for discussions on $T_{coating}$.

The size range mentioned here (i.e., 130−160 nm) is for the rBC core, rather than for the whole rBC containing particle. After accounting for coating, the size range of the whole rBC

containing particle was at least ~ 170 nm, recalling that rBC coating thicknesses varied between ~ 20−40 nm for the different flight screens during F_9/4. We also investigated the coating thicknesses determined for larger rBC cores (i.e., the rBC cores in the $D_{MEV}$ range of 160−190 nm; $D_{MEV}$ indicates the mass equivalent diameter of the rBC core). As shown in the figure above, evolutions of coating thickness exhibited the same pattern for rBC cores in the $D_{MEV}$ ranges of 130−160 and 160−190 nm. We prefer to present coating thickness results for rBC cores in the $D_{MEV}$ range of 130−160 nm, mainly because the fraction of rBC cores that can be assigned a coating thickness ($F_{assigned}$) was the highest for this $D_{MEV}$ range (Figure 11 in the revised manuscript).

In addition, we also examined the counts of rBC cores that can be assigned a coating thickness for the $D_{MEV}$ ranges of 130−160 and 160−190 nm, i.e., $N_{145}$ and $N_{175}$ (145 and 175 correspond to the centers of these two $D_{MEV}$ bins). $N_{175}$ were found to be much smaller than $N_{145}$. For example, the $N_{145}$ to $N_{175}$ ratios were ~ 3−5 for successive flight screens of F_9/4; and moreover, $N_{175}$ were as low as ~ 100 for the out-of-plume conditions during F_9/4. We think that ~ 100 rBC containing particles are too few for a quantitative analysis of coating thickness. This is other reason why we prefer to use 130−160 nm rBC cores for $T_{coating}$ discussion.

The discussions above were reflected in the revised manuscript.

**(25)** L352. The authors cannot simultaneously argue for more "aged" air outside the plume (and supported by $NO_x/NO_y$) and non-OS local emissions. Or, if they are going to do so, they need to make a more concrete and persuasive argument here in my opinion.

**Our response:** Only the argument on non-OS local emissions was kept in the revised manuscript.

**(26)** L356. Why would the authors not compare in/out of plume $T^*$ values based on their $NO_x/NO_y$, rather than the "screen"? A lot of this discussion would benefit from a more direct link to photochemical age. Right now, the concept of photochemical age is seemingly left behind to earlier discussions, but it also belongs here in my opinion.

**Our response:** The dependence of rBC coating thickness on photochemical age will be presented elsewhere, together with quantitative discussions on the evolution of non-rBC containing particles (as mentioned in our response to the second specific comment raised by the second Referee).

**(27)** L376. In this discussion, the authors should note more explicitly that they are working from the small-size side of things, compared to these other studies. Generally, one might expect

LEO to be more robust for larger particles, with larger signal. Of course, there is a limitation because as particles get too large the scattering detector is saturated and a full Gaussian cannot be fit. Overall, the point that there is uncertainty on the order of 10% in the optical diameter from the LEO fit is worth reporting, but the authors should note the issue that they are using small particles.

**Our response:** The uncertainties reported here were derived from all the detected non-rBC containing particles, including both relatively large and small ones. Nonetheless, we agree with the reviewer that the LEO fit should be more robust for larger particles. We also agree with the reviewer that "small particles" were used in this manuscript for the discussions on coating thickness, which can potentially lead to biases in interpretation. As mentioned in our response to Comment# 24, evolutions of coating thickness exhibited the same pattern for relatively "small" and "large" rBC cores, indicating that our interpretation of coating thickness and the related conclusions should be valid. The discussions above were reflected in the revised manuscript.

**(28)** Fig. 11b/L363. While true, the authors neglect that the OA/rBC is smaller for the in-plume conditions overall. While not all OA will be coated on rBC, that the OA/rBC is so much smaller in plume might lead one to think that the coating amount on the rBC in plume should be smaller compared to out of plume, opposite what is reported. This should be discussed, in my opinion.

**Our response:** The following sentences were added as suggested: "It should also be noted that the out-of-plume OA are dominated by pre-existing secondary organic aerosols formed from biogenic precursors (Liggio et al., 2016), which do not contribute to the formation of coating materials on rBC cores. This explains why the out-of-plume conditions have higher OA/rBC ratios but in general lower $T^*$ compared to in plumes.".

**References**

Buffaloe, G. M., Lack, D. A., Williams, E. J., Coffman, D., Hayden, K. L., Lerner, B. M., Li, S. M., Nuaaman, I., Massoli, P., Onasch, T. B., Quinn, P. K., and Cappa, C. D.: Black carbon emissions from in-use ships: a California regional assessment, Atmos. Chem. Phys., 14, 1881–1896, 2014.

Cappa, C. D., Onasch, T. B., Massoli, P., Worsnop, D. R., Bates, T. S., Cross, E. S., Davidovits, P., Hakala, J., Hayden, K. L., Jobson, B. T., Kolesar, K. R., Lack, D. A., Lerner, B. M., Li, S. M., Mellon, D., Nuaaman, I., Olfert, J. S., Petäjä, T., Quinn, P. K., Song, C., Subramanian, R., Williams, E. J., and Zaveri, R. A.: Radiative absorption enhancements due to the mixing state of atmospheric black carbon, Science, 337, 1078–1081, 2012.

Kondo, Y., Matsui, H., Moteki, N., Sahu, L., Takegawa, N., Kajino, M., Zhao, Y., Cubison, M. J., Jimenez, J. L., Vay, S., Diskin, G. S., Anderson, B., Wisthaler, A., Mikoviny, T., Fuelberg, H. E., Blake, D. R., Huey, G., Weinheimer, A. J., Knapp, D. J., and Brune, W. H.: Emissions of black carbon, organic, and inorganic aerosols from biomass burning in North America and Asia in 2008, J. Geophys. Res., 116, D08204, doi:10.1029/2010JD015152, 2011.

Laborde, M., Crippa, M., Tritscher, T., Jurányi, Z., Decarlo, P. F., Temime-Roussel, B., Marchand, N., Eckhardt, S., Stohl, A., Baltensperger, U., Prévôt, A. S. H., Weingartner, E., and Gysel, M.: Black carbon physical properties and mixing state in the European megacity Paris, Atmos. Chem. Phys., 13, 5831–5856, 2013.

Laborde, M., Mertes, P., Zieger, P., Dommen, J., Baltensperger, U., and Gysel, M.: Sensitivity of the Single Particle Soot Photometer to different black carbon types, Atmos. Meas. Tech., 5, 1031–1043, 2012.

Liggio, J., Gordon, M., Smallwood, G., Li, S. M., Stroud, C., Staebler, R., Lu, G., Lee, P., Taylor, B., and Brook, J. R.: Are emissions of black carbon from gasoline vehicles underestimated? Insights from near and on-road measurements, Environ. Sci. Technol., 46, 4819–4828, 2012.

Liggio, J., Li, S. M., Hayden, K., Taha, Y. M., Stroud, C., Darlington, A., Drollette, B. D., Gordon, M., Lee, P., Liu, P., Leithead, A., Moussa, S. G., Wang, D., O'Brien, J., Mittermeier, R. L., Brook, J. R., Lu, G., Staebler, R. M., Han, Y., Tokarek, T. W., Osthoff, H. D., Makar, P. A., Zhang, J., Plata, D. L., and Gentner, D. R.: Oil sands operations as a large source of secondary organic aerosols, Nature, 534, 91–94, 2016.

McMeeking, G. R., Hamburger, T., Liu, D., Flynn, M., Morgan, W. T., Northway, M., Highwood, E. J., Krejci, R., Allan, J. D., Minikin, A., and Coe, H.: Black carbon measurements in the boundary layer over western and northern Europe, Atmos. Chem. Phys., 10, 9393–9414, 2010.

Moteki, N., Kondo, Y., Miyazaki, Y., Takegawa, N., Komazaki, Y., Kurata, G., Shirai, T., Blake, D. R., Miyakawa, T., and Koike, M.: Evolution of mixing state of black carbon particles: aircraft measurements over the western Pacific in March 2004, Geophys. Res. Lett., 34, L11803, doi:10.1029/2006GL028943, 2007.

Schwarz, J. P., Spackman, J. R., Fahey, D. W., Gao, R. S., Lohmann, U., Stier, P., Watts, L. A., Thomson, D. S., Lack, D. A., Pfister, L., Mahoney, M. J., Baumgardner, D., Wilson, J. C., and Reeves, J. M.: Coatings and their enhancement of black carbon light absorption in the tropical atmosphere, J. Geophys. Res., 113, D03203, doi:10.1029/2007JD009042, 2008.

---

## Author Response (AR2)

Dear Editor,

Heading of section 2.4 has been corrected as suggested. Thanks!

Sincerely yours,

Yuan Cheng, Ph.D.

Environment and Climate Change Canada